

# Contribution of personal weather stations to the observation of deep-convection features near the ground

Marc Mandement and Olivier Caumont

CNRM, Université de Toulouse, Météo-France, CNRS, Toulouse, France

**Correspondence:** Marc Mandement (marc.mandement@meteo.fr)

**Abstract.** The lack of observations near the surface is often cited as a limiting factor in the observation and prediction of deep convection. Recently, networks of personal weather stations (PWSs) measuring pressure, temperature and humidity in near-real time have been rapidly developing. Even if they suffer from quality issues, their high temporal resolution and their higher spatial density than standard weather station (SWS) networks have aroused interest in using them to observe deep convection.

In this study, the PWSs contribution to the observation of deep-convection features near the ground is evaluated. Four cases of deep convection in 2018 over France were considered using data from Netatmo, a PWS manufacturer. A fully automatic PWS processing algorithm, including PWS quality control, was developed. After processing, the mean number of observations available increased by a factor of 134 in mean sea level pressure (MSLP), of 11 in temperature and of 14 in relative humidity over the areas of study. Near-surface SWS analyses, and analyses comprising standard and personal weather stations (SPWS)

were built. The usefulness of crowdsourced data was proven both objectively and subjectively for deep convection observation. Objective validations of SWS and SPWS analyses by leave-one-out cross-validation (LOOCV) were performed using SWSs as the validation dataset. Over the four cases, LOOCV root mean square errors (RMSEs) decreased for all parameters in SPWS analyses compared to SWS analyses. RMSEs decreased by 73 to 77 % in MSLP, 12 to 23 % in temperature and 17 to 21 % in relative humidity. Subjectively, fine-scale structures showed up in SPWS analyses, while partly or not at all visible in SWS

observations only. MSLP jumps accompanying squall lines or individual cells were observed, as well as wake lows at the rear of these lines. Temperature drops and humidity rises accompanying most of the storms were observed sooner and at a finer resolution in SPWS analyses than in SWS analyses. The virtual potential temperature was spatialized at an unprecedented spatial resolution. It gave the opportunity to observe cold pool propagation and secondary convective initiation over areas with high virtual potential temperatures, i.e. favorable locations for near surface parcel lifting.

## 1 Introduction

The increasing number of connected objects – i.e. with Internet access – which carry meteorological sensors has raised the interest of scientists because they are a supplementary means of observing the atmosphere. Several publications emphasize the high potential of these sensors for the fine-scale observation of atmospheric phenomena, in complement with traditional sources, given the unprecedented spatio-temporal resolution of the networks constituted by these sensors (Muller et al., 2015;

Chapman et al., 2017). These new observations come from smartphones (Overeem et al., 2013; Mass and Madaus, 2014;



McNicholas and Mass, 2018a), connected vehicles (Mahoney and O'Sullivan, 2013) or personal weather stations (PWSs here-after, also called citizen weather stations) (Bell et al., 2013) for example. All these studies underline the potential gains in the fine-scale description of the atmosphere near the ground.

Among all meteorological processes, deep convection induces the sharpest variations of pressure, temperature, humidity,
wind and rain near the ground. Recognition of deep-convection surface features such as low-level convergence boundaries (Wilson and Schreiber, 1986; Wakimoto and Murphey, 2010), pressure, temperature and humidity features prior to convection (Madaus and Hakim, 2016) as well as tracking their temporal evolution is deemed crucial for thunderstorm forecasting (Sanders and Doswell, 1995). The prediction of convective initiation location and timing, as well as its evolution still remain a challenging question. Several studies agree that there is a lack of observations at the surface, which limits the quality of
analyses at the convective scale which in turn limits good forecasts of convective events (Stensrud and Fritsch, 1994; Fowle and Roebber, 2003; Snook et al., 2015; Sobash and Stensrud, 2015). Observational studies highlight a need of additional high resolution observations (Adams-Selin and Johnson, 2010; Clark, 2011), because deep convection is often associated with small-scale parameters variations. Also there is a need to validate high-resolution numerical models and verify their accuracy of deep convection modelling.

These statements motivated experiments in several fields of meteorology. They have been led using denser observational net-works near the ground for the study of a storm in particular, for urban climate studies or for hydrological applications. Several assimilation experiments of denser weather stations networks or observations made with connected objects have also already been performed. Madaus et al. (2014) performed an hourly assimilation of dense pressure observations from mesonets. Results showed increase in short-term forecast accuracy for temperature, wind and pressure near mesoscale phenomena. Sobash and
Stensrud (2015); Gasperoni et al. (2018) showed that 5-min assimilation of mesonet data, mostly thermodynamic observations, improved forecasts of convection initiation. Regarding connected objects, several recent data assimilation experiments focused on smartphone observations (Madaus and Mass, 2017; McNicholas and Mass, 2018b; Hintz et al., 2019). Assimilating smartphone pressure measurements leaded to some improvements in analyses and short-term forecasts of surface variables compared to experiments without assimilation of observations. Results shown were strongly sensitive to the quality control
techniques developed in each study. It demonstrates that quantifying the uncertainties associated to these observations, and establish robust quality control procedures is crucial. In parallel, recent work has been done in the urban climate communities that work about phenomena at a city scale, benefiting from a high number of connected objects due to the high population density in cities. Temperature measurements from PWSs have been used to visualize the urban heat island in several Western Europe cities (Wolters and Brandsma, 2012; Chapman et al., 2017; Meier et al., 2017; Napoly et al., 2018). These studies
showed that PWSs can provide robust estimates of temperature at a fine scale when measurements are quality-controlled and spatially aggregated. For precipitation, de Vos et al. (2017, 2018) showed that rain gauges produced by Netatmo, a PWS manufacturer, can be used for urban rainfall monitoring, capturing well small-scale rainfall dynamics in Amsterdam. Recently, Clark et al. (2018) showed the benefit of PWS data in the life-cycle observation of a hailstorm that crossed the United Kingdom in 2015. The supplementary data in temperature, pressure, wind speed given by the PWSs associated with the UK Met Office net-





**Table 1.** Periods of time of each case study.

| Date | 26 May 2018 | 4 July 2018 | 15 July 2018 | 28 August 2018 |
|---|---|---|---|---|
| Hour of beginning (UTC) | 06:55 | 05:55 | 00:15 | 12:55 |
| Hour of end (UTC) | 23:55 | 21:55 | 23:55 | 23:55 |

work revealed fine-scale structures corresponding to conceptual models of severe thunderstorms. However, the quality control procedures were not fully automatic and a manual check of each dataset had to be performed.

The goal of the present study is to evaluate the contribution of PWSs to the existing standard weather station (SWS) network in the observation of deep-convection processes at midlatitudes, focusing over France. A fully automatic PWS processing

algorithm based on comparisons with SWSs was developed. The features near the ground of isolated storms, multicellular systems or supercell storms are observed, extending the work of Clark et al. (2018) which focused on a sole supercell storm. Observed features of processes responsible for their formation or generated by these systems such as cold pools, gust fronts and sea breeze effects are studied. In order to do so, mean sea level pressure (MSLP), temperature and humidity gridded analyses including PWSs are built. The additional value of these weather stations is objectively evaluated by comparison with reference

gridded analyses made only with SWSs. First, in Sect. 1 this study describes interesting convective cases of the spring and summer 2018 over France. In Sect. 2, a presentation of the different weather station networks used in the study is made. The processing including a quality control of PWSs measurements is detailed in Sect. 3, followed by the validation performed against SWSs in Sect. 4. Then, a focus on some features observed during the different convective cases is made in Sect. 5 to evidence the positive contribution of PWSs.

## 2   Overview of the cases

Four cases are chosen to evaluate the contribution of PWS network to the observation of deep convection features near the ground. The considered cases and the periods of time of the cases are indicated in Table 1.

### 2.1   26 May 2018: bow echo over the west of France

On 26 May 2018 a midlevel low at 500 hPa was located in the Bay of Biscay (Fig. 1). It induced a moderate southerly flux over

France: the Bordeaux sounding at 23:00 UTC 25 May observed a $11\,\mathrm{m\,s^{-1}}$ southerly wind at 500 hPa. Two positive (cyclonic) upper-level potential vorticity anomalies circulated during the day in the southerly flux observed near the tropopause (Fig. 2a). At the surface, a shallow pressure low around 1010 hPa in the Bay headed north very slowly during the day. Pressure gradients were weak all over the western part of France. Over the south-west of France, the air was mild and humid due to the convective activity that occurred the day before and the early hours of the 26. Indeed, a first mesoscale convective system (MCS) evolved

mainly on the Atlantic ocean, its edges affecting the French Atlantic coast from the Basque Country to Brittany between

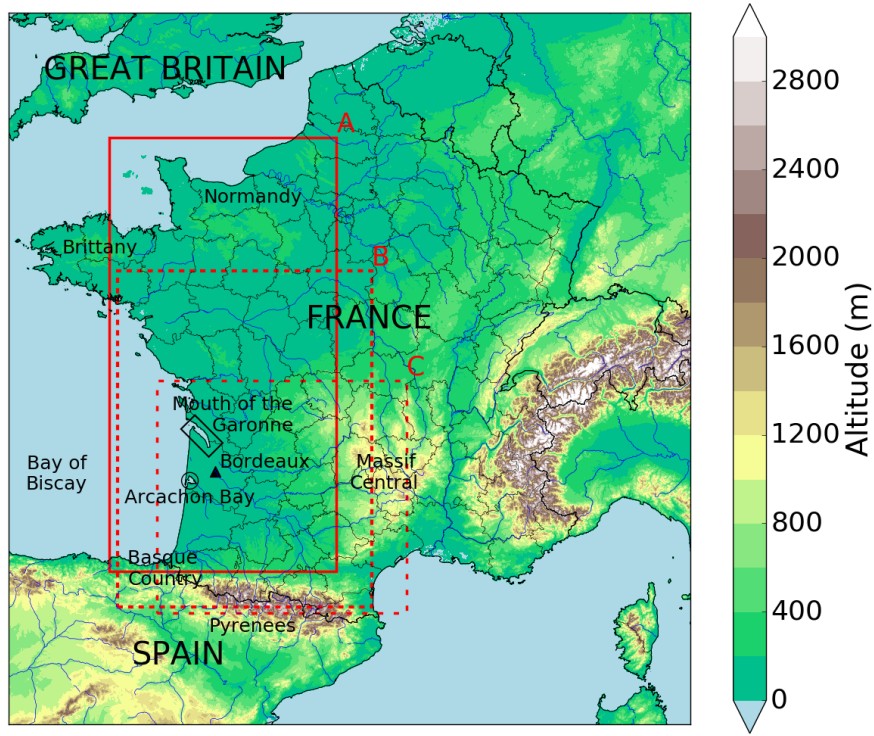

**Figure 1.** Topographic map of Metropolitan France. Domains studied are drawn in red and locations mentioned are indicated in black. Domain A corresponds to the 26 May case, domain B corresponds to the 28 August case and domain C corresponds to the 4 July and the 15 July cases.

the 25 May 22:00 UTC and 26 May 16:00 UTC. The Bordeaux 25 May 23:00 UTC sounding, the closest available before the event, exhibited only 416 $\mathrm{J\,kg^{-1}}$ surface-based convective available potential energy (SBCAPE) and strong 245 $\mathrm{J\,kg^{-1}}$ convective inhibition (CIN) because of the decrease of temperature near the surface due to the diurnal cycle. But it recorded a 1583 $\mathrm{J\,kg^{-1}}$ most unstable CAPE (MUCAPE) from the lifting of a 435 m AGL air parcel (25 $\mathrm{J\,kg^{-1}}$ CIN for this parcel). It

5 shows the presence of unstable levels above the stable nocturnal boundary layer. The sounding observed a moderate 0–6 km above ground level (AGL) 16 $\mathrm{m\,s^{-1}}$ wind shear and the hodograph exhibited a clockwise rotation of the winds in the 0–1 km AGL layer resulting in a 69 $\mathrm{m^2\,s^{-2}}$ helicity.

At the rear of the first MCS, thunderstorms formed in the north of Spain, west of the Pyrenees between 06:00 UTC and 08:00 UTC. These cells, advected by the mid-troposphere southern flux crossed the Pyrenees mountains and headed north

10 towards Bordeaux. A squall line organization of the thunderstorms appeared around 10:00 UTC. This MCS transitioned into a bow echo around 13:00 UTC according to radar reflectivities and crossed the west of France moving in a south-north orientation from the Bordeaux region towards Normandy and Great Britain (Fig. 3a). The system was still active when it left the French territory at 23:00 UTC. The path followed by the bow echo can be seen west of France in Fig. 4a, marked by the area of high



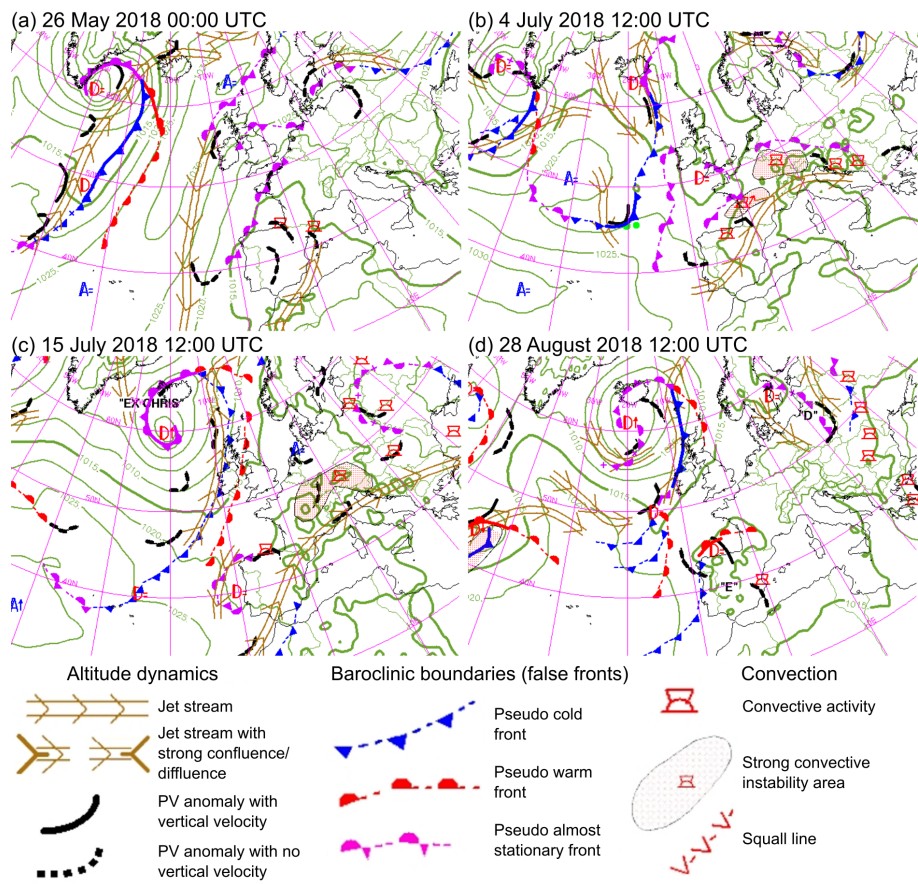

**Figure 2.** Europe and North Atlantic graphical synoptic analysis charts (a) at 00:00 UTC 26 May 2018, (b) at 12:00 UTC 4 July 2018, (c) at 12:00 UTC 15 July 2018 and (d) at 12:00 UTC 28 August 2018 (Santurette and Joly, 2002). MSLP isobars from ARPEGE model 6 h forecast of the T-6 h run (T: time of the chart) are drawn at 5 hPa intervals. Surface fronts are shown by conventional symbols. Lows are indicated by "D" and highs by "A" with the pressure tendency observed. Adapted from Météo-France national forecast department.

speed, diverging wind gusts. Also, to the north of the MCS, a severe isolated thunderstorm, identified as a supercell in the radar imagery, developed around 10:30 UTC and merged with the MCS around 15:00 UTC.

The supercell produced damaging hail up to 4 cm in diameter and rain up to 22 mm in 6 min in the center of Bordeaux. The bow echo produced mainly strong wind gusts up to 31 m s$^{-1}$: 13 SWSs recorded gusts higher than 25 m s$^{-1}$ and 4 of them gusts higher than 28 m s$^{-1}$ (Fig. 4a). The two systems resulted in one fatality, 1 555 rescue operations and 10 000 homes without power. It produced also local flash floods in Bordeaux and hail damages in the Bordeaux vineyards.
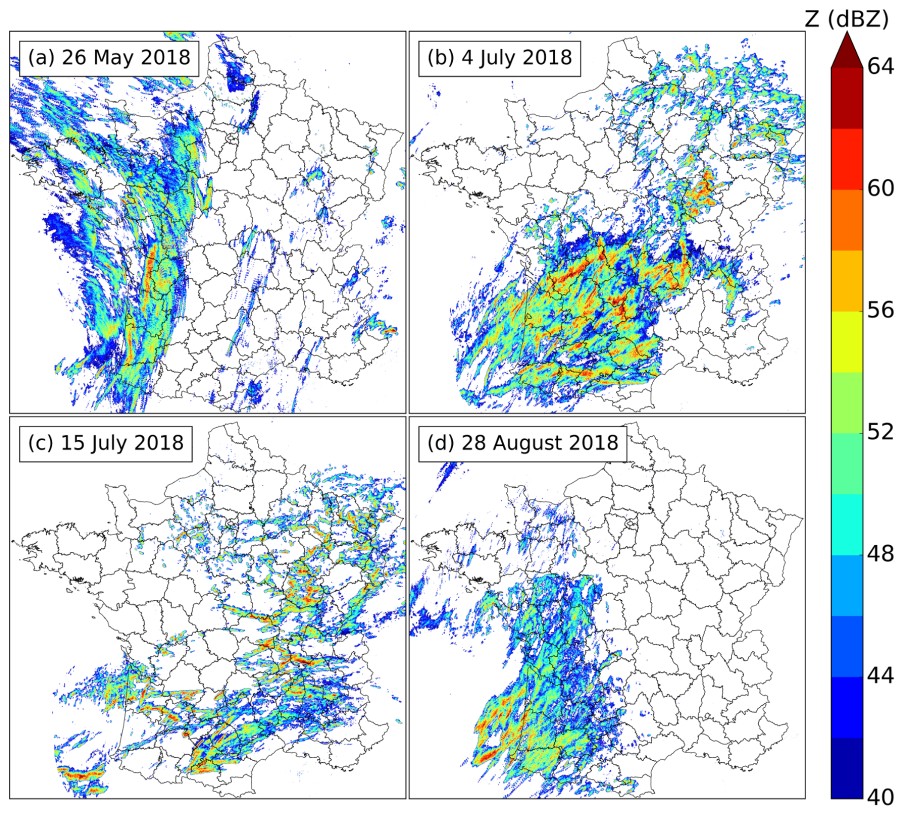

**Figure 3.** Maximum base reflectivity for each case study: (a) 26 May 2018, (b) 4 July 2018, (c) 15 July 2018 and (d) 28 August 2018.

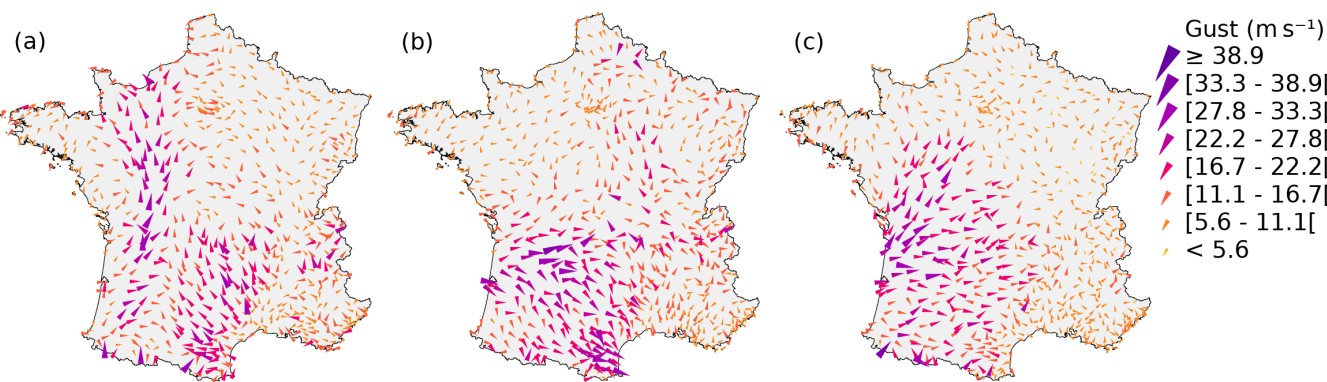

**Figure 4.** Peak wind gusts measured over metropolitan France (a) on 26 May 2018, (b) on 4 July 2018 and (c) on 28 August 2018. Adapted from Météo-France climatological service.



## 2.2 4 July 2018: squall line over the south-west of France

On 4 July 2018, a midlevel low at 500 hPa was located over the Atlantic ocean and was extended by a trough across the Iberic peninsula. The trough moved west towards France during the day, inducing a moderate south-westerly flux at midlevel over the west of France: the Bordeaux sounding observed a southwesterly $12 \, \mathrm{m \, s^{-1}}$ wind at 23:00 UTC 3 July and south-southwesterly

$16 \, \mathrm{m \, s^{-1}}$ wind at 11:00 UTC 4 July. An upper-level potential vorticity anomaly circulated during the afternoon over France at the rear of the convective area and on the left side of a jet stream branch (Fig. 2b). At the surface, a shallow low was located south-west of England and pressure gradients were weak all over France. Isolated thunderstorms affected the south-west of France on the night of 3 to 4 July. In the morning of the 4 July, around 09:00 UTC, numerous thunderstorms developed in the north of Spain, over the Bay of Biscay and in the south-west of France. They aggregated in several multicellular systems.

Embedded in one of these systems over the south-west of France, one of the storms exhibited supercellular characteristics in the radar imagery. Around 12:00 UTC, another multicellular system formed north of Spain, strengthened over the Atlantic ocean and transitioned into a squall line. The squall line headed north-east while isolated storms formed in its southern part. It finally merged with other storms around 17:00 UTC and the isolated cells south of it merged in clusters, most of them evolving along with a strong MSLP gradient area (Fig. 3b). The sounding of Bordeaux at 11:00 UTC, before the arrival of the squall line at

14:00 UTC, exhibited large $2155 \, \mathrm{J \, kg^{-1}}$ SBCAPE, weak $12 \, \mathrm{J \, kg^{-1}}$ CIN and a moderate clockwise rotating wind hodograph, resulting in 0–3 km AGL $79 \, \mathrm{m^2 \, s^{-2}}$ helicity. The system generated peak wind gusts up to $34 \, \mathrm{m \, s^{-1}}$. A large area was affected by strong wind gusts: more than 30 SWSs recorded gust speed higher than $25 \, \mathrm{m \, s^{-1}}$ and 11 of them gusts higher than $28 \, \mathrm{m \, s^{-1}}$ (Fig. 4b). Flash floods were observed with rain rates up to 41.6 mm in 18 min. It resulted in one fatality, six injuries, 2 500 rescue operations and 185 000 homes without power. Tennis ball-sized hail (>6 cm in diameter) was reported: in a village

named Saint-Sornin, 800 houses were seriously damaged. This hail was caused by the storm identified as a supercell, in which reflectivities up to 70 dBZ were measured by radar.

## 2.3 15 July 2018: isolated storms over the south-west of France

On 15 July 2018, a midlevel trough at 500 hPa circulated from Portugal towards west of France inducing westerly-to-southwesterly winds in mid-troposphere. An upper-level potential vorticity anomaly circulated over the south-west of France

during the afternoon in a north-eastward direction (Fig. 2c). At the surface, pressure gradients were weak over France. The sounding of Bordeaux at 11:00 UTC exhibited a SBCAPE of $1790 \, \mathrm{J \, kg^{-1}}$ and a CIN of $0 \, \mathrm{J \, kg^{-1}}$, showing ideal conditions for the development of surface-based convection. A sea breeze established near the Atlantic shore and its effects on cloud coverage were visible on satellite images at 12:39 UTC (Fig. 5a). A frontier appeared between the coastal band where temperatures reached 27 to 30 °C with clear sky, and the inland area where temperatures reached 32 to 33.5 °C and cumulus clouds

were developing. Surface observations and satellite images showed the wind convergence due to the breeze moving eastwards between 12:30 UTC and 13:45 UTC. Around 13:10 UTC, towering cumuli turned into cumulonimbi at the south-east of the Arcachon Bay (Fig. 5b) where SWSs measured the strongest temperature gradient with 5 °C difference in 40 km distance. The initiation happened along the wind convergence line. The first cell triggered secondary cell development west and north




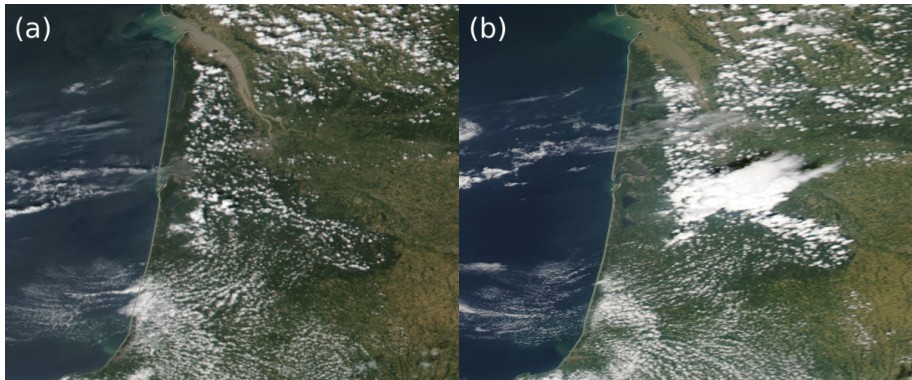

**Figure 5.** Visible satellite images of convective initiation due to sea breeze convergence on 15 July 2018 taken by (a) Suomi NPP/VIIRS at 12:39 UTC and (b) Aqua/MODIS at 13:43 UTC. Images from NASA Worldview.

of it. Two main cells split and evolved in different directions: the first one headed north-northeast and the other one headed east (Fig. 3c). The cell evolving north-northeast caused high wind gusts up to $34 \, \mathrm{m \, s^{-1}}$ at Bordeaux airport at 15:05 UTC, and temperature dropped by $15.5 \, \mathrm{°C}$ in 23 minutes. Hail diameter up to $3 \, \mathrm{cm}$ was observed in the Bordeaux region under the thunderstorm. This event caused 84 rescue operations and happened in a context of public gatherings due to the football world
5  cup final.

## 2.4  28 August 2018: two squall lines over the west of France

On 28 August 2018, a midlevel trough at $500 \, \mathrm{hPa}$ concerned the west of France and moved slightly eastwards during the day, resulting in a south-southeasterly flux. At the surface a low centered north of the Iberic peninsula deepened during the day (Fig. 2d), helped by a potential vorticity anomaly at upper levels and located left of the jet, near a diffluent exit region visible
10  at 18:00 UTC (not shown). The sounding of Bordeaux at 11:00 UTC exhibited a SBCAPE of $1283 \, \mathrm{J \, kg^{-1}}$ and a strong CIN of $300 \, \mathrm{J \, kg^{-1}}$. The hodograph showed a strong unidirectional 0–6 km AGL wind shear reaching $25 \, \mathrm{m \, s^{-1}}$. Thunderstorms formed south of the low i.e. over sea and in the north of Spain; they crossed the Pyrenees and the Bay of Biscay between 15:00 UTC and 17:00 UTC and reached French southwestern territory between 17:00 UTC and 18:00 UTC as multicellular systems. The northern part of the MCS evolved in squall line between 18:00 UTC and 19:00 UTC while the southern part
15  formed a second squall line at the rear (Fig. 3d). The two lines generated gusts up to $31 \, \mathrm{m \, s^{-1}}$; 15 SWSs recorded wind gusts higher than $25 \, \mathrm{m \, s^{-1}}$ and 6 of them recorded gusts higher than $28 \, \mathrm{m \, s^{-1}}$ (Fig. 4c). It resulted in two people slightly injured, $10\,000$ homes without power, around 200 rescue operations and 9 forest fires generated by lightning. Hail up to $8 \, \mathrm{mm}$ in diameter was reported near the coast.





**Table 2.** Maximum available sensors of SWSs and PWSs, i.e. emitting at least one measurement, during the case studies over Metropolitan France.

| Number of sensors (% of stations equipped) | SWS | PWS |
| --- | --- | --- |
| Temperature | 1 032 (100 %) | 36 473 (83 %) |
| Precipitation | 1 005 (97 %) | 11 912 (27 %) |
| Wind | 736 (71 %) | 5 763 (13 %) |
| Relative humidity | 705 (68 %) | 36 472 (83 %) |
| Surface pressure | 192 (19 %) | 42 029 (95 %) |
| Number of weather stations | 1 032 | 44 115 |

## 3 Datasets

Two main surface networks are used: automatic SWSs taken as a reference and Netatmo PWSs. To associate surface features to the thunderstorm activity, storms are mainly tracked with the French radar network.

### 3.1 SWS network

SWSs are all automatic Météo-France operational weather stations sampling atmospheric parameters at a time step of 1 min. These weather stations have been installed, maintained and quality-controlled by Météo-France. The requirements in term of accuracy for Météo-France weather stations are $\pm$ 0.5 °C in temperature and $\pm$ 6 % in relative humidity (Tardieu and Leroy, 2003). They are taken as a reference in this study. The maximum number of weather stations measuring each physical parameter during the cases of 2018 is shown in Table 2. The SWS least measured parameter over France is surface pressure with only 19 % of SWSs equipped. The number of humidity and wind sensors equipping SWSs is respectively 3.7 to 3.8 times larger than the number of pressure sensors. Also, there are 5.4 and 5.2 times as many temperature and precipitation sensors as pressure sensors. Additional automatic weather stations, owned by Météo-France or its partners with only 5 min, 6 min or hourly measurements are not part of the SWS dataset, but are used for verification. It represents approximately 800 stations measuring temperature and 250 measuring relative humidity.

### 3.2 PWS network

A PWS dataset made of all Netatmo automatic weather stations available over Metropolitan France is used. During the case studies of 2018, a maximum of 44 115 different PWSs recorded at least one observation which is approximately 15 times the total number of professional automatic weather stations currently available at Météo-France. Among these PWSs, 95 % recorded pressure measurements, 83 % temperature and relative humidity measurements, 27 % rain measurements and 13 % wind measurements. The 15 July for example, PWSs provided a total of 5 625 137 surface pressure observations, 4 837 133 temperature observations and 4 836 843 relative humidity observations during the case study.



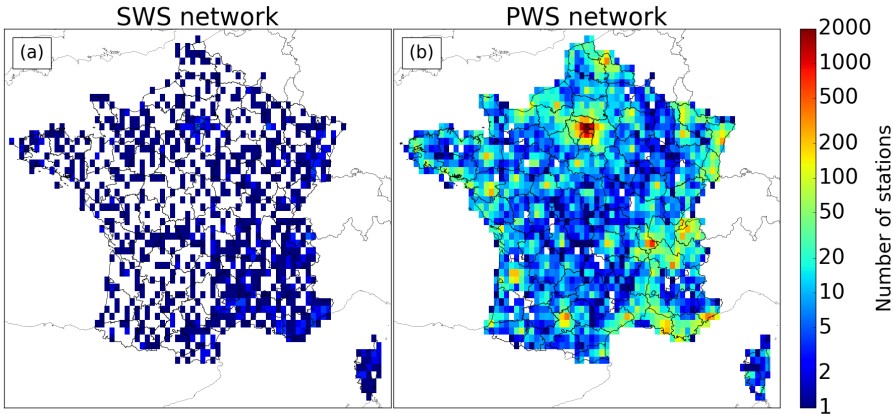

**Figure 6.** Number of SWSs (a) and PWSs (b) over Metropolitan France on 4 July 2018. Observation counts are binned into approximately $0.2° \times 0.2°$ bins.

The metadata associated with each station is quite basic: a unique identification number, the latitude, the longitude and the altitude. The altitude of 17 % of PWSs is missing. During the year 2017, the number of PWSs recording at least once in a month increased from around 37 800 in January to around 44 000 in December, showing the rapid development of this network.

The transmission of data by these PWSs is based on radio waves between outdoor and indoor modules, on Wi-Fi between the indoor module and the personal internet box, and then by different methods but essentially wires between the personal internet box and the internet service provider. At each step, technical failures or user-related shutdowns can occur. In each file transmitted by the PWS's manufacturer, 10 to 15 % of the total number of PWSs are not providing measurements. It can be due to disconnection between station modules, disconnection of the personal internet box, power or internet outages.

PWS measurements are irregular in time whereas meteorological networks are usually designed to perform them at regular time steps. The mean time step between two measurements is 5 min, but PWS owners can also perform on-demand measurements. Netatmo provided in near real time only 10 min time step measurements, which is the minimum time step used in this study. On average, most of the measurements are done at the minutes 5, 15, 25, 35, 45 and 55 of each hour. Also, the mean spacing between PWSs is not regular whereas the average separation of SWSs is about 30 km. The spatial density of PWSs is highly correlated to the population density (Fig. 6).

## 3.3 Radar

The operational weather radar network between May and August 2018 in Metropolitan France is composed of 30 radars. Five radars in the south of France are S-band radars, twenty radars are C-band, and there are five X-band radars. In this study, the French operational base reflectivity, i.e. measured at the lowest elevation angle of the radar, mosaicked from these 30 radars is used. It has a $1 \times 1$ km$^2$ spatial resolution and a 5-min temporal resolution with reflectivities ranging from –9 to 70 dBZ with a 0.5 dBZ step. For every pixel in the mosaic, the maximum base reflectivity from radars distant by 180 km or less is taken. If





the pixel is distant by more than 180 km to every radar, the maximum reflectivity of radars at a distance between 180 km and 250 km is taken. More details on French radar network are given by Figueras i Ventura and Tabary (2013).

## 4   Data processing

To compare PWS and SWS time series, a linear interpolation of each PWS time series was done at the minutes 5, 15, 25, 35,
45 and 55 of each hour because most of the measurements are done at these times. These interpolated time series are referred to as raw PWS time series.

The inspection of raw PWS time series for all parameters shows major departures compared to SWS time series, which confirms the necessity of a quality control as already stated in previous studies (Bell et al., 2013; Muller et al., 2015; Meier et al., 2017; Napoly et al., 2018). Measurements provided by PWSs have a lot of uncertainties due to heterogeneous and
unknown environmental conditions. The ground type, the direct exposure of PWS sensors to solar radiation or heat sources, the lack of ventilation, the lack of maintenance or calibration problems can lead to errors. Field tests realized at Météo-France with 3 Netatmo PWSs show errors in temperature of about $0\,°C \pm 0.9\,°C$ in median and 95 % interval compared to a reference SWS, and errors in relative humidity of about $+3\,\% \pm 7\,\%$ in median and 95 % interval. These tests have been performed with a supplementary radiation shield: they show a correct quality of temperature and humidity sensors when properly protected but
do not give insights about their accuracy without this shield. For pressure, some sources of errors exposed by McNicholas and Mass (2018a) in smartphone pressure sensors apply to PWS pressure sensors because they are similar microelectromechanical systems (MEMS). According to their study, errors result from different response time of sensors to pressure changes, sensor bias, inaccurate metadata or user-related issues (pressurized environments, below or above ground level PWS locations). The STMicroelectronics MEMS pressure sensor mounted on Netatmo PWS has a $\pm\,1$ hPa absolute accuracy (Netatmo, 2019).
Because of the uncertainties affecting PWS measurements and the departures observed in comparison to reference measurements, an automatic PWS data processing algorithm was built. It includes a quality control in pressure, temperature and humidity which is designed to be simple and efficient whatever the meteorological situation is. The algorithm is mainly based on comparisons with a quality-controlled reference network as it was done by Meier et al. (2017) and Clark et al. (2018). The data processing is performed during the periods of time indicated in Table 1. Cases begin before convection initiation and end
after convection dissipation of the storm systems studied over the area of interest. In order to accurately evaluate PWSs and be able to detect abnormal behaviour, calm conditions are necessary during most of the time. Indeed, if storms affect weather stations at each time step, conclusions about the quality of the measurements by comparing it to a reference or close stations may be dubious, given the small scale of some phenomena.

### 4.1   Gridding methods

For temperature, relative humidity, MSLP and surface pressure, all gridded analyses derived from observations are built at a 10-min time step and $0.01°$ resolution in latitude and longitude ($\approx 1.1$ km N/S and $\approx 0.8$ km E/W at $45°$ N) by interpolating, for each grid point, weather stations available in the vicinity. The gridding method used is the inverse distance weighting (IDW)




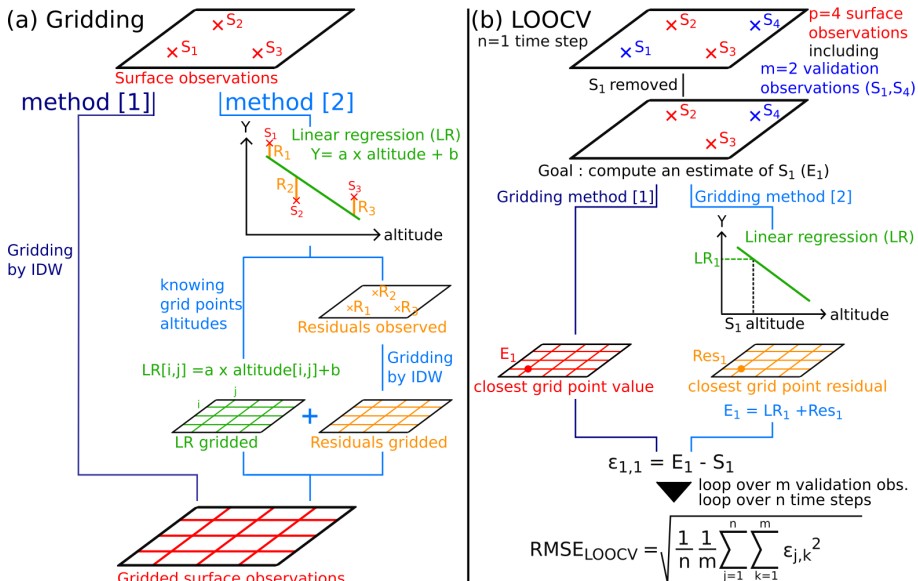

**Figure 7.** (a) Gridding methods used to build analyses from discrete surface observations. MSLP and relative humidity are gridded by the method [1] while temperature and surface pressure are gridded by the method [2]. (b) LOOCV algorithm explained through the case of four observations made by four weather stations, including two validation stations during one time step. The complete LOOCV is a loop performed over all time steps ($n$) and over all validation stations chosen ($m$). The loop provides an array of errors $(\epsilon_{j,k})_{1 \leq j \leq n, 1 \leq k \leq m}$ of dimension $n \times m$, allowing to compute a RMSE over $n \times m$ observations or a RMSE associated to a validation station only over $n$ observations. If the estimate is equal to the observation, error is equal to zero.

with a power factor of two. Weather stations too far away from a grid point are not used in the computation. For temperature and relative humidity, SWSs distant by more than 60 km are not taken into account ; this radius is set to 30 km for PWSs. For MSLP and surface pressure, SWSs distant by more than 100 km are not taken into account ; this radius is also set to 100 km for PWSs. The radius is larger for pressure because it is the minimum radius allowing to cover the entire Metropolitan France.

5   A maximum of 10 SWSs and 30 PWSs are used at each grid point in the IDW.

MSLP and relative humidity are directly gridded (Fig. 7a method [1]). For temperature and surface pressure, because they vary strongly with altitude, a linear regression with respect to the altitude is performed first. After that, the residuals (i.e. the difference between the values obtained by linear regression and the observations) are gridded as shown in Fig. 7a method [2], and then added to the grid derived from the linear regression. The linear regression uses only the SWS observations used in the

10   gridding. For temperature, the linear regression is adapted: to diminish the predominant weight of the low altitude SWSs over the highest SWSs, SWSs are binned in vertical layers of 100 m height. The mean temperature and the mean altitude of SWSs comprised in each layer are computed. A linear regression is then performed over these vertical layers. This choice was made to be closest to the observed temperature lapse rate rather than using a constant lapse rate.

The reference analyses called SWS analyses used in the following sections are built only with SWS data.





## 4.2 Computation of PWS MSLP

Even when the altitude of the Netatmo PWS ($z_{\text{PWS}}$) is unknown, the PWS still provides a pressure value. In fact, under the name of pressure Netatmo provides two different quantities:

- a MSLP ($\text{MSLP}_{\text{PWS}}$) computed from the hydrostatic equation assuming a constant 15 °C temperature and a 0 % relative
humidity at sea level if $z_{\text{PWS}}$ is known (83 % of cases)

$$\text{MSLP}_{\text{PWS}} = P \left( 1 - \frac{\Gamma z_{\text{PWS}}}{T_0} \right)^{-\frac{gM}{\Gamma R_0}} \tag{1}$$

where $P$ is the surface pressure measured at the PWS in hPa, $T_0 = 288.15$ K is the sea level temperature of the International Civil Aviation Organization (ICAO) standard atmosphere, $\Gamma = 0.0065$ K m$^{-1}$ is the ICAO environmental lapse rate in the troposphere below 11 km, $g = 9.80665$ m s$^{-2}$ is the standard acceleration of gravity, $M = 0.0289644$ kg mol$^{-1}$
is the molar mass of dry air, and $R_0 = 8.31447$ J mol$^{-1}$ K$^{-1}$ is the ideal gas constant.

- surface pressure $P$ if $z_{\text{PWS}}$ is unknown (17 % of cases)

To compare $\text{MSLP}_{\text{PWS}}$ to SWS measurements, it was necessary to recalculate the MSLP. The formula used to calculate the MSLP for SWSs is the one in use at Météo-France and is the same as that used by, e.g. Garratt (1984). It takes into account the observed surface temperature and humidity at the weather station:

$$\text{MSLP} = P \exp \left( \frac{gMz}{R_0 \overline{T_v}} \right) = P \exp \left( \frac{\frac{gM}{R_0} z}{T_v + \frac{\Gamma}{2} z} \right) = P \exp \left( \frac{0.03414z}{T_v + 0.00325z} \right) \tag{2}$$

with $\overline{T_v}$ the mean virtual temperature in the fictitious air column extending from sea-level to the level of the station, which is equal to $T_v + \frac{\Gamma}{2} z$ considering the decrease of the virtual temperature with altitude at a constant lapse rate $\Gamma$ in this column.

The virtual temperature $T_v$ at the weather station is derived from $T$, the 2-m temperature in Kelvin ($t$ is $T$ in degrees Celsius), the 2-m water vapour pressure $e = \frac{U}{100} e_w$ in hPa where $U$ is the 2-m relative humidity in %, and $e_w$ is the saturation
water vapour pressure in hPa obtained through World Meteorological Organization (2012) formula. $T_v$ and $e_w$ are computed as follows:

$$T_v = \frac{T}{1 - \frac{0.378 e}{P}} = \frac{T}{1 - \frac{0.378 U e_w}{100 P}} \quad \text{with} \quad e_w = 6.112 \exp \left( \frac{17.62 t}{t + 243.12} \right) \tag{3}$$

$T$ and $U$ are derived from the nearest point of the SWS analyses. The altitude $z$ is equal to $z_{\text{PWS}}$ if the difference in altitude is less than 15 m between $z_{\text{PWS}}$ and the Shuttle Radar Topography Mission (SRTM) digital elevation model (DEM) extracted
from Python package "altitude" (Tom de Ruijter, 2016). If the difference is larger than 15 m, the DEM altitude is taken. The value is chosen to keep the benefit of accurate altitudes that may be given by internal GPS of smartphones to the Netatmo mobile application during the PWS set up process. It results in more accurate altitude when the PWS is located in a small building for example. Then, comparing metadata to a DEM eliminates altitude errors that may be introduced by users : they may erroneously modify PWS altitude because it is a way to modify the value of PWS pressure.



### 4.3 PWS systematic error correction

The motivation to compare Netatmo measurements to SWS analyses is to eliminate systematic errors. Some of them are due to the PWS itself such as sensor quality or the impossibility of maintenance by design; some are due to the environmental conditions where the PWS is set up, but some are due to PWS owners who can calibrate sensors as they wish. The mobile

phone application allows users to calibrate the temperature sensor and modify the altitude, which has an influence on pressure. All sensors can be calibrated by personal requests to Netatmo.

For relative humidity, PWS time series are compared to the SWS analyses at the closest grid point. For surface pressure and temperature, because they vary rapidly with altitude, PWS time series are not compared directly to the SWS analyses at the closest grid point. Indeed, the altitude of the PWS closest grid point may be really different of the PWS actual one. That

is why a more precise calculation is performed: the altitude $z$ defined previously considered as the closest to the actual PWS altitude is used in the computation. Residuals time series are taken from the closest grid point residuals. This precise calculation corresponds to SWS analyses having an accurate ground altitude at PWS locations.

For each PWS, the median of the errors between the time series derived from SWS analyses at its location ($\boldsymbol{x_a}$) and its raw PWS time series ($\boldsymbol{x_r}$) is obtained. The corrected PWS time series ($\boldsymbol{x_c}$) is computed by removing the median of the errors from

the raw PWS time series.

$$\boldsymbol{x_c} = \boldsymbol{x_r} - \text{med}(\boldsymbol{x_r} - \boldsymbol{x_a})\boldsymbol{l} \tag{4}$$

with $\boldsymbol{x_c}, \boldsymbol{x_r}, \boldsymbol{x_a}, \boldsymbol{l}$ column vectors gathering a single PWS time series are of dimension $n$ equal to the number of time steps of a case. Here, $\boldsymbol{l} = \{1, \ldots, 1\}$.

The choice of the median is explained by the observation of large variations in temperature, humidity or pressure due to deep

convection. Because of the lower density of the SWS network compared to the PWS network, some of these variations that are actual signals affect the calculation of mean error. Using the median allows to ignore a major part of these physical deviations while identifying systematic errors affecting PWSs. This procedure is close to the one followed by Madaus et al. (2014) which is performed during periods of several months.

In the following parts, all PWS time series refer to corrected PWS time series. The steps leading to these PWS corrected

time series, i.e. the computation of PWS MSLP and the PWS systematic error correction are referred to as PWS preprocessing.

### 4.4 PWS data quality control

Two common filters are applied to pressure, temperature and humidity. A PWS is removed if it has the same latitude and longitude as another, and less than half of the measurements are available. For the computation of MSLP, PWSs with altitude higher than 750 m are discarded, as recommended by the World Meteorological Organization (2012). Then a last filter is

applied in order to discard PWSs that are not providing accurate measurements.

For temperature and relative humidity, the last filter is based on the assumption that the larger the differences between PWS time series and SWS analyses during the case study and the longer they last, the less confidence is put in PWS measurements.



For each PWS, the root mean square error (RMSE) of PWSs temperature and relative humidity time series ($x_c$) against time series derived from SWS analyses ($x_a$) is computed. It is hereafter called RMSE$_{PWS}$, with $n$ the number of time steps:

$$\text{RMSE}_{\text{PWS}} = \sqrt{\frac{1}{n} \sum_{j=1}^{n} (x_c[j] - x_a[j])^2} \qquad (5)$$

The filter eliminates PWSs with RMSE$_{PWS}$ higher than an adaptive threshold called RMSE$_{thresh}$:

$$\text{RMSE}_{\text{PWS}} > \text{RMSE}_{\text{thresh}} \qquad (6)$$

To determine the RMSE$_{thresh}$, an automatic algorithm based on leave-one-out cross-validation (LOOCV, see Fig. 7b) was built. Consider $p$ surface stations (PWSs and SWSs) producing observations including $m$ validation stations (SWSs only, $p \geq m$). For a given time step $j \in [1; n]$, the LOOCV removes one validation observation $k \in [1; m]$. Using $p - 1$ observations (all except the observation $k$), an estimate at the removed observation location $E_k(p)$ is computed through the gridding method described in Sect. 4.1. Then, the estimate is compared to the actual observation $S_k$, giving an error $\epsilon_{j,k}(p)$:

$$\epsilon_{j,k}(p) = E_k(p) - S_k \qquad (7)$$

The process is reproduced over the $m$ validation stations and the $n$ time steps of the case study, giving an array of $m \times n$ errors, from which the LOOCV RMSE is computed:

$$\text{RMSE}_{\text{LOOCV}}(p) = \sqrt{\frac{1}{n} \frac{1}{m} \sum_{j=1}^{n} \sum_{k=1}^{m} \epsilon_{j,k}(p)^2} \qquad (8)$$

The lower the errors, the closer to the observations the estimates are. Thereby, RMSE$_{LOOCV}(p)$ can be chosen as a metric evaluating the accuracy of the $p$ surface stations from which the estimates are built.

Let $x$ be the unknown RMSE$_{thresh}$. Then, $p(x)$ is the number of PWSs and SWSs which verify RMSE$_{PWS} \leq x$ ($m$ is the number of SWSs). The RMSE$_{thresh}$ chosen is the $x$ that minimizes RMSE$_{LOOCV}(p(x))$:

$$\text{RMSE}_{\text{thresh}} = \underset{x}{\text{argmin}} \, \text{RMSE}_{\text{LOOCV}}(p(x)) \qquad (9)$$

For large values of $x$, $p(x)$ tends to the total number of PWSs and SWSs remaining after the two common filters, and so RMSE$_{LOOCV}(p(x))$ tends to large values, because almost all PWSs are kept including those exhibiting abnormal behaviours. For small values of $x$, $p(x)$ tends to $m$, the number of SWSs, and RMSE$_{LOOCV}(p(x))$ tends to quite large values because of the small number of SWSs and their large spacing.

The resulting RMSE$_{thresh}$ picked up by the algorithm depends on the case, varying from 1.10 °C to 1.45 °C in temperature 25   and from 5.5 % to 7.5 % in relative humidity.

For MSLP and surface pressure, instead of a threshold, PWSs providing suspicious measurements are eliminated one by one by an algorithm. It consists in a LOOCV using SWSs and PWSs as validation stations ($m = p$) that eliminates one suspicious


**Table 3.** Number of PWS filtered at each step of the quality control in temperature (T) and relative humidity (U) over the area of each case study.

| Case study | 26 May 2018 | | 4 July 2018 | | 15 July 2018 | | 28 August 2018 | |
|---|---|---|---|---|---|---|---|---|
| | T | U | T | U | T | U | T | U |
| Number of PWS time series | 11 372 | | 5 113 | | 5 063 | | 7 347 | |
| Same latitude/longitude | 100 | | 37 | | 32 | | 48 | |
| >50 % missing values | 615 | 616 | 324 | 326 | 296 | 298 | 448 | 448 |
| $\text{RMSE}_{\text{PWS}} > \text{RMSE}_{\text{thresh}}$ | 6 731 | 6 242 | 2 508 | 2 141 | 3 103 | 2 947 | 4 051 | 4 735 |
| $\text{RMSE}_{\text{thresh}}$ | 1.10 °C | 6.5 % | 1.40 °C | 7.5 % | 1.45 °C | 7.5 % | 1.20 °C | 5.5 % |
| PWS remaining | 3 926 | 4 414 | 2 244 | 2 609 | 1 632 | 1 786 | 2 800 | 2 116 |
| % of total PWS | 35 % | 39 % | 44 % | 51 % | 32 % | 35 % | 38 % | 29 % |

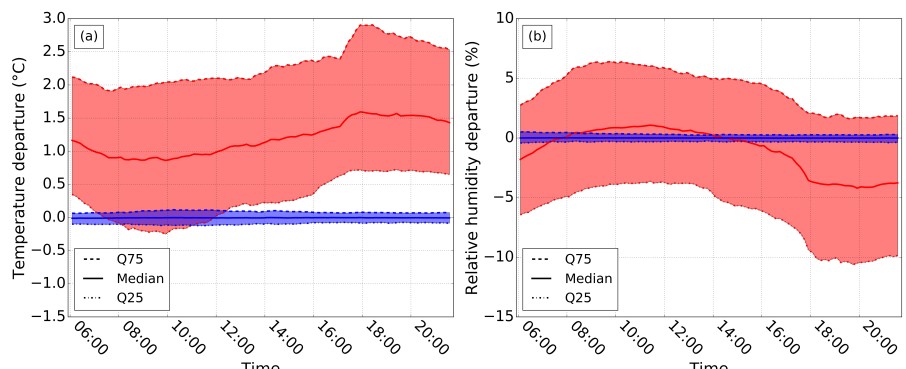

**Figure 8.** Illustration of the PWS processing in (a) temperature and (b) relative humidity for the 4 July 2018 case. The distribution of departures between PWS measurements and their corresponding SWS analysis values every 10 min is shown with the quantile 25 (Q25), the median, and the quantile 75 (Q75). In red, raw PWS time series; in blue: processed PWS time series.

PWS at each step $s$. PWSs are used in the validation dataset this time because SWS coverage is quite sparse. A one by one elimination is possible because only few PWS errors remain after the first three filters in pressure. The suspicious PWS is identified by computing the RMSE associated with all validation stations $k$, $k \in [1; m]$ which is:

$$\text{RMSE}_{\text{LOOCV,k}}(p) = \sqrt{\frac{1}{n} \sum_{j=1}^{n} \epsilon_{j,k}(p)^2} \qquad (10)$$

5    The PWS with the highest $\text{RMSE}_{\text{LOOCV,k}}(p)$ is physically the one which disagrees the most in RMSE with all neighbour PWSs and SWSs during the case study, which is suspicious. This station is eliminated. The algorithm stops when $\text{RMSE}_{\text{LOOCV}}(p)$ increases at step $s+1$ compared to step $s$. Physically, an increase means that a PWS which was in strong agreement with at





**Table 4.** Number of PWSs filtered in MSLP at each step of the quality control for each case study.

| Case study | 26 May 2018 | 4 July 2018 | 15 July 2018 | 28 August 2018 |
|---|---|---|---|---|
| Number of PWS time series | 13 098 | 5 820 | 5 783 | 8 432 |
| Identical latitude/longitude | 107 | 41 | 36 | 56 |
| >50 % missing values | 523 | 316 | 277 | 431 |
| Altitude >750 m | 7 | 175 | 165 | 105 |
| LOOCV removal algorithm | 155 | 81 | 80 | 65 |
| PWS remaining (% of PWS time series) | 12 306 (94 %) | 5 207 (89 %) | 5 225 (90 %) | 7 775 (92 %) |

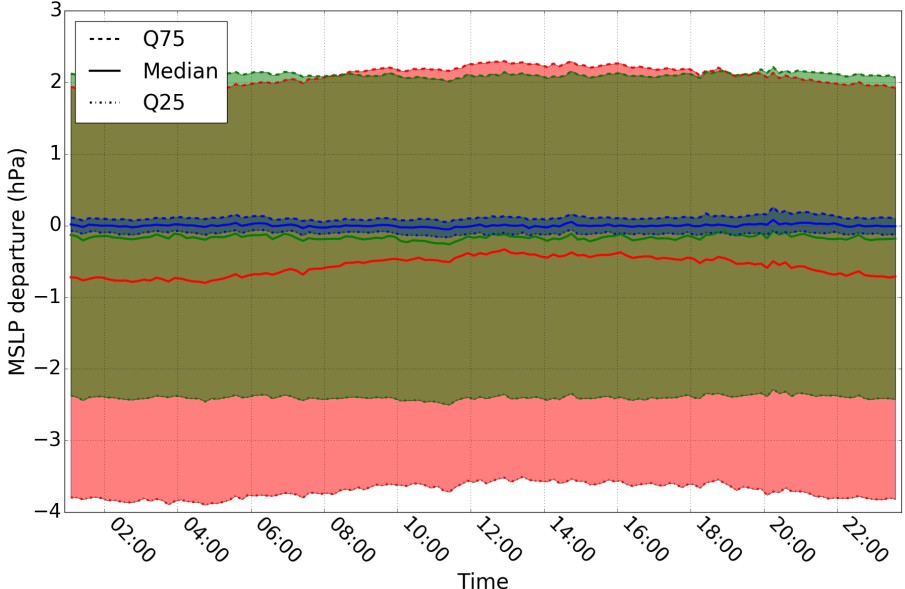

**Figure 9.** Illustration of the PWS processing in MSLP for the 26 May 2018 case. The distribution of departures between PWS measurements and their corresponding SWS analysis values every 10 min is shown with the Q25, the median, and the Q75. In red, raw PWS time series; in green: preprocessed PWS time series; in blue: processed PWS time series.

least one neighbour station (PWS or SWS, called $k'$) was eliminated at step $s$. At step $s+1$, $k'$ captures some physical process (local low or high) but is alone in doing it, and so $\text{RMSE}_{\text{LOOCV,k'}}(p)$ increases, as well as the RMSE of some PWS around it. As a consequence, the resulting $\text{RMSE}_{\text{LOOCV}}(p)$ taking into account all PWS contributions increases. This algorithm is well fitted for pressure because most of the errors affecting PWSs are uncorrelated, and few PWSs provide erroneous values. It will

5 probably not work for other parameters like temperature whose errors may be spatially correlated (errors because of direct radiation for example). Each step of quality control in MSLP is detailed in Table 4.



The result of PWS processing is illustrated for temperature in Fig. 8a, for relative humidity in Fig. 8b and for MSLP in Fig. 9. PWS measurements are compared at different time steps to the SWS analyses before and after processing. In temperature (Fig. 8a), the distribution of departures before processing exhibits systematic positive departures with a diurnal cycle. The daily minimum of the median departures is reached in the morning between 08:00 UTC and 10:00 UTC, after sunrise and the

daily maximum is reached in the evening or the night between 17:00 UTC and 06:00 UTC, in the 4 July case but also in the other cases not shown. In relative humidity (Fig. 8b), the distribution of departures before processing also exhibits a diurnal cycle with positive departures during the day and negative departures during the night, in all cases. In MSLP (Fig. 9), the distribution of departures before processing seems to exhibit a small diurnal cycle, with departures increasing in the morning and decreasing in the evening. For all parameters, the processing shifts the distribution of departures near zero and strongly

decreases the width of the interquartile range of departures. This shows the efficiency of the algorithm in diminishing departures to SWS analyses while keeping features associated with deep convection, as it was designed for (see Sect. 6).

## 5   Validation

After PWS time series were processed, the remaining PWSs were combined to SWSs. This network is called hereafter SPWS network, and the gridded fields produced with this network are called SPWS analyses. The additional value of SPWS analyses

compared to SWS analyses is evaluated quantitatively in MSLP, temperature and relative humidity. Also, for temperature and relative humidity, in order to evaluate the role of processing in the results obtained, raw PWS time series and SWS time series were combined. The network associated to this dataset is called SPWS_raw network. It is not done for pressure because the raw dataset blends MSLP and surface pressure as explained in Sect. 4.2.

LOOCVs are performed on SWS, SPWS and SPWS_raw observations ($p$ observations) and validated on SWS observations

($m$ observations) included in these datasets. The median of $\epsilon_{j,k}(p)$ over all validation stations $k \in [1;m]$ and all time steps $j \in [1;n]$ is computed. The $\mathrm{RMSE_{LOOCV}}(p)$ abbreviated in RMS is also shown. The mean and quartiles of $\epsilon_{j,k}(p)$, not shown in the tables, have also been scrutinized. All experiments are compared to the SWS experiment.

### 5.1   MSLP

In MSLP (Table 5), a decrease ranging from 0.01 hPa to 0.05 hPa in absolute value of the median error is observed in

SPWS experiments compared to SWS experiments, depending on the case. The absolute value of the mean error decreases in three cases and remains stable in one case: it is less than 0.02 hPa for all cases in SPWS experiments. A decrease ranging from 0.32 hPa to 0.48 hPa in the interquartile range of errors is observed for all cases in SPWS experiments compared to SWS experiments. Also, very substantial decrease in RMSE reaching 73 % to 77 % are observed. These results quantitatively show that adding PWS measurements in observed MSLP analyses strongly improves their accuracy. For MSLP, the number of

available observations is multiplied by 134 in mean over the four cases with the SPWS network compared to the SWS network.



**Table 5.** Statistics of the LOOCV performed on SWS and SPWS observations of MSLP and validated on SWS observations for each case study. The evolution in % is relative to the RMSE of SWS observations.

| Case study | | 26 May 2018 | | 4 July 2018 | | 15 July 2018 | | 28 August 2018 | |
|---|---|---|---|---|---|---|---|---|---|
| Network | | SWS | SPWS | SWS | SPWS | SWS | SPWS | SWS | SPWS |
| MSLP error (hPa) | Median | 0.012 | –0.001 | 0.015 | –0.002 | 0.030 | –0.002 | 0.048 | 0.002 |
| | RMS | 0.404 | 0.099 | 0.702 | 0.187 | 0.449 | 0.104 | 0.611 | 0.151 |
| | % of evolution | | **–75** % | | **–73** % | | **–77** % | | **–75** % |

**Table 6.** Statistics of the LOOCV performed on SWS, SPWS_raw and SPWS observations of temperature and validated on SWS observations for each case study. The evolution in % is relative to the RMSE of SWS observations.

| Case study | | 26 May 2018 | | | 4 July 2018 | | | 15 July 2018 | | | 28 August 2018 | | |
|---|---|---|---|---|---|---|---|---|---|---|---|---|---|
| Network | | SWS | SPWS _raw | SPWS | SWS | SPWS _raw | SPWS | SWS | SPWS _raw | SPWS | SWS | SPWS _raw | SPWS |
| Temperature error (°C) | Median | 0.009 | 1.122 | 0.005 | 0.013 | 1.386 | 0.008 | –0.025 | 1.289 | –0.017 | –0.050 | 0.725 | –0.008 |
| | RMS | 1.060 | 1.823 | 0.889 | 1.272 | 2.178 | 1.124 | 1.518 | 2.137 | 1.258 | 1.333 | 1.875 | 1.028 |
| | % of evolution | | **+72** % | **–16** % | | **+71** % | **–12** % | | **+41** % | **–17** % | | **+41** % | **–23** % |

## 5.2 Temperature

In temperature (Table 6), a positive shift of the median error ranging from 0.73 °C to 1.39 °C is observed in SPWS_raw experiments compared to SWS experiments. Bias reaches 0.74 °C to 1.45 °C and increase in RMSE ranges from 41 % to 72 % compared to SWS experiments. These results show the key role of processing: without this step, adding PWSs strongly
5  decreases the quality of analyses.

For SPWS experiments, a decrease ranging from 0.00 °C to 0.04 °C in absolute value of the median error is observed compared to SWS experiments, depending on the case. The absolute value of the mean error shows no particular trend and remains less than 0.07 °C for all cases in SPWS experiments. It indicates that PWSs do not introduce substantial bias or shifts in the temperature distribution. A decrease ranging from 0.06 °C to 0.22 °C in the interquartile range of errors is observed for
10  all cases in SPWS experiments compared to SWS experiments. Also, substantial decrease in RMSE reaching 12 % to 23 % are observed. These results quantitatively show that adding PWS measurements in temperature analyses improves their accuracies. For temperature, the number of available observations is multiplied by 11 in mean over the four cases with the SPWS network compared to the SWS network.





**Table 7.** Statistics of the LOOCV performed on SWS, SPWS_raw and SPWS observations of relative humidity and validated on SWS observations for each case study. The evolution in % is relative to the RMSE of SWS observations.

| Case study | | 26 May 2018 | | | 4 July 2018 | | | 15 July 2018 | | | 28 August 2018 | | |
|---|---|---|---|---|---|---|---|---|---|---|---|---|---|
| Network | | SWS | SPWS _raw | SPWS | SWS | SPWS _raw | SPWS | SWS | SPWS _raw | SPWS | SWS | SPWS _raw | SPWS |
| Relative humidity error (%) | Median | –0.238 | –3.268 | –0.170 | 0.004 | –1.163 | –0.204 | 0.125 | –0.616 | –0.090 | 0.203 | 2.753 | –0.027 |
| | RMS | 6.820 | 8.898 | 5.375 | 7.667 | 8.864 | 6.364 | 9.567 | 10.093 | 7.605 | 7.482 | 9.360 | 5.920 |
| | % of evolution | | **+31** % | **–21** % | | **+16** % | **–17** % | | **+6** % | **–21** % | | **+25** % | **–21** % |

## 5.3 Relative humidity

In relative humidity (Table 7), shifts of the median error ranging from –3.3 % to 2.7 % are observed in SPWS_raw experiments compared to SWS experiments. Biases reach –2.3 % to 1.9 % and RMSEs increase range from 6 % to 31 % compared to SWS experiments. These results show the key role of processing: without this step, adding PWSs strongly decreases the quality of analyses.

5    For SPWS experiments, the absolute value of the median error is less than or equal to 0.2 % and the absolute value of the mean error remains less than 0.6 %. It indicates that PWSs do not introduce any substantial bias or shifts in the relative humidity distribution. Decrease ranging from 0.0 % to 1.9 % in the interquartile range of errors is observed for all cases in SPWS experiments compared to SWS experiments. Also, substantial decrease in RMSE reaching 17 % to 21 % is observed. These results quantitatively show that adding PWS measurements in relative humidity analyses improves their accuracies. For relative humidity, the number of available observations is multiplied by 14 in mean over the four cases with the SPWS network compared to the SWS network.

## 6    Results for selected convective cases

In the following section comparisons are made between SWS and SPWS networks by showing observed values at station
15  locations or by comparing SWS and SPWS analyses.

### 6.1    Contribution of PWSs to MSLP analyses

#### 6.1.1    26 May 2018

At 12:45 UTC 26 May 2018, a squall line was located over the south-west of France. The MSLP field of SWS analysis (Fig. 11a) exhibits a single pressure high reaching 1014.9 hPa, in the western part of the MCS, south of the highest reflectivities. It does
20  not show significant pressure perturbations or strong pressure gradients in the eastern part of the MCS. On SPWS analysis (Fig. 11b), a crescent-shaped pressure high associated with the system is identified with MSLP reaching 1015.0 to 1015.5 hPa,


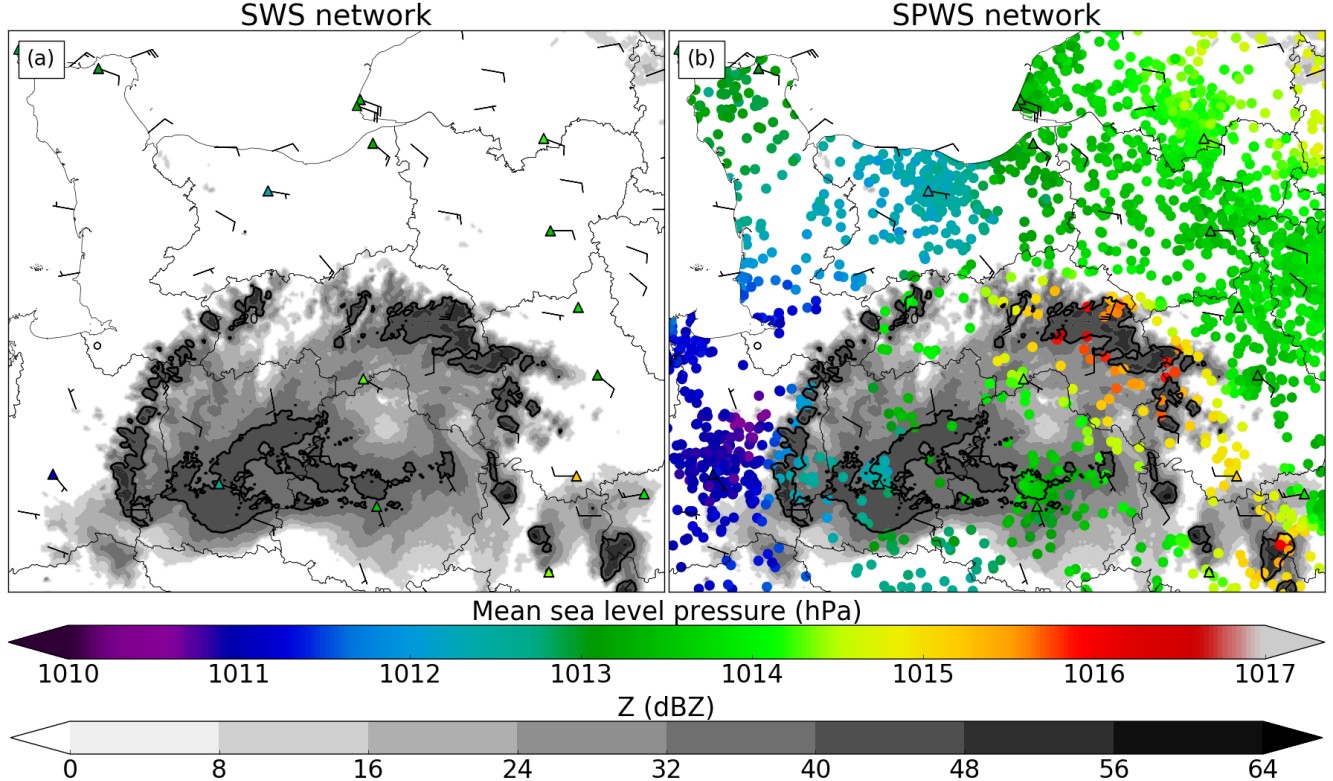

**Figure 10.** MSLP observations of (a) SWS network and (b) SPWS network at 18:15 UTC 26 May. SWSs are indicated by coloured triangles with black contours and PWSs by coloured circles. The instantaneous wind gust is shown with barbs. Base reflectivity (Z) in grey colours indicates thunderstorm activity and location. Reflectivities over 40 dBZ are materialized by bold black contours.

especially near and under the highest reflectivities in the convective part of the storm. A MSLP low is located at the rear of the stratiform part. Along the strong pressure gradients revealed by the SPWS network, high wind gusts of $19\,\mathrm{m\,s^{-1}}$ at 12:10 UTC and $25\,\mathrm{m\,s^{-1}}$ at 12:38 UTC were observed in the eastern part of the MCS. The MSLP field agreeing the most with MSLP anomalies described by the theory of squall lines (Johnson and Hamilton, 1988; Haertel and Johnson, 2000) is found in SPWS

5   analysis. SPWS analysis is also more coherent with surface wind observations than SWS analysis. A rise in MSLP under the supercell evolving ahead of the squall line is also exhibited by the SPWS analysis, but is not observed in the SWS analysis whereas MSLP rise is usually observed under supercellular storms (Clark et al., 2018). Effects of this cell near the ground are confirmed by a nearby SWS which recorded a $22\,\mathrm{m\,s^{-1}}$ gust. An interesting MSLP feature is shown by SPWS analysis in this case: MSLP field exhibits a crescent-shaped structure from 12:25 UTC whereas the same structure is observed in reflectivity

10  only from 13:00 UTC. The feature is observed just before the squall line evolve in a bow echo.

At 18:45 UTC, major differences between both analyses appear in MSLP. A surge in pressure associated with the bow echo is not visible in the SWS analysis (Fig. 11c) while the SPWS analysis shows it (Fig. 11d). At the surface high winds were


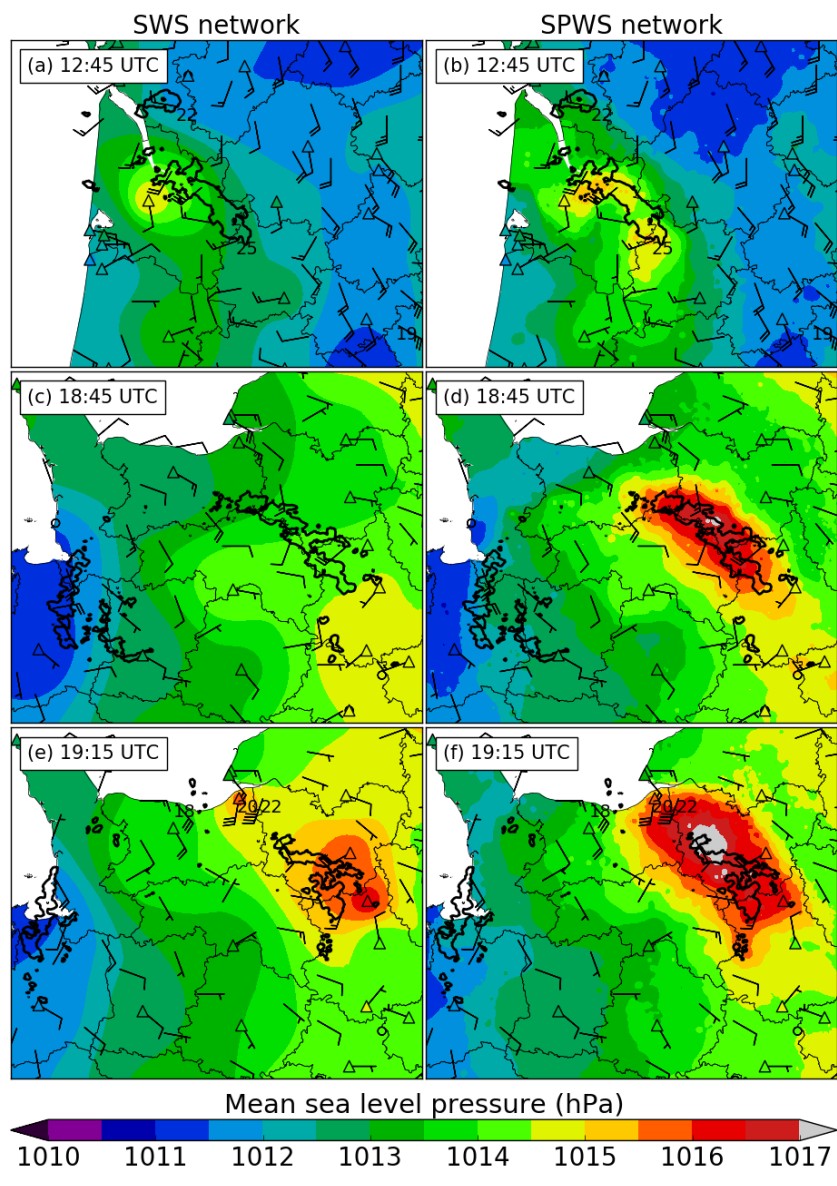

**Figure 11.** MSLP (a,c,e) SWS analyses and (b,d,f) SPWS analyses at (a,b) 12:45 UTC, (c,d) 18:45 UTC and (e,f) 19:15 UTC 26 May. Reflectivities over 40 dBZ are materialized by bold black contours. SWS MSLP measurements are shown by coloured triangles with black contours. The instantaneous wind gust is shown with barbs, and the highest gust during the last 10 min over 17 m s⁻¹ is annotated nearby.

observed: a SWS recorded a 20 m s$^{-1}$ wind gust under the gust front at 18:48 UTC. Moreover, the high exhibited by the SPWS analysis is collocated with reflectivities over 40 dBZ indicating the location of thunderstorm cores. These clues, associated with the brutal increase in MSLP observed further by SWSs agree with the presence of a MSLP high, absent of the SWS analysis.





At 19:15 UTC (Figs. 11e,f), the bow echo was over Normandy. SPWS analysis exhibits a pressure surge associated with the convective system with strong gradients of MSLP, especially in the part of the bow echo perpendicular to the propagation direction. This feature is coherent with the 18:45 UTC SPWS analysis and confirmed by a SWS on its path: it recorded a surge in MSLP reaching 1.5 hPa in 10 min and 2.8 hPa in 1 h. The MSLP pressure front observed in the SPWS analysis is only

partially seen by the SWS analysis: the SWS analysis exhibits independent MSLP surges and misses the MSLP maximum exhibited by the SPWS analysis. Moreover, at the same time, the radar network observed a decrease of reflectivities, especially in the northern part of the MCS, indicating a decay of the convective activity. However, near the surface, the SPWS analysis does not exhibit the vanishing of the MSLP high associated with the gust front. This decay is neither observed in surface winds: high wind gusts collocated and temporally synchronized with the gust front described by SPWS analyses were still observed. Gusts

were recorded by 4 SWSs located in the north of Normandy, near the sea: $20 \, \mathrm{m \, s^{-1}}$ at 19:11 UTC, $22 \, \mathrm{m \, s^{-1}}$ at 19:13 UTC, $25 \, \mathrm{m \, s^{-1}}$ at 19:25 UTC and $23 \, \mathrm{m \, s^{-1}}$ at 19:27 UTC. Also, in the western part of the bow echo, where few thunderstorms remained, SPWS analysis shows weaker MSLP gradients than in the northern part of the bow echo. It is confirmed by SWSs that observed only moderate wind gusts.

The SWS network alone is not able to seize most MSLP features associated with this MCS, exhibited by SPWS network.

Wind speed, wind direction as well as SWS MSLP measurements are temporally and spatially coherent with the SPWS analyses, strengthening the confidence in this new analysis. In similar cases, the indication of remaining sharp MSLP gradients while radar reflectivities are decaying may help forecasters to keep warning about possible strong gusts near the surface, which seems relevant given the gusts observed in this case. During the bow echo life cycle, the SPWS analyses exhibit several pressure surges as described by Adams-Selin and Johnson (2010). SPWS network may be used in further studies to compare these

observed surges to their pressure surge-new bowing cycle theory.

### 6.1.2    4 July 2018

At 13:55 UTC 4 July 2018, a well-formed squall line located near the mouth of the Garonne river was moving towards northeast. It generated a west-northwest $25 \, \mathrm{m \, s^{-1}}$ gust in Bordeaux at 13:49 UTC and a west-southwest $18 \, \mathrm{m \, s^{-1}}$ gust at 13:55 UTC in another SWS. The SWS analysis in Fig. 12a exhibits a MSLP high, but pressure gradients remain moderate: there is no

indication of a strengthening wind in the area. In the SPWS analysis visible in Fig. 12b, sharp gradients of MSLP appear at the location of the observed strong gusts. The SPWS analysis is coherent with the location of convective cells indicated by radars, with the surges in MSLP measured by SWSs, and also with the wind gust directions almost perpendicular to MSLP gradients observed in the southern part of the line.

At 15:35 UTC, the SPWS analysis in Fig. 12d exhibits strong MSLP gradients while SWS analysis in Fig. 12c shows weak

gradients in comparison. These strong MSLP gradients are coherent with the position of the convective cells and the measured surface gust speeds. Gusts up to $32 \, \mathrm{m \, s^{-1}}$ at 15:26 UTC and $31 \, \mathrm{m \, s^{-1}}$ at 15:32 UTC are recorded by SWSs.



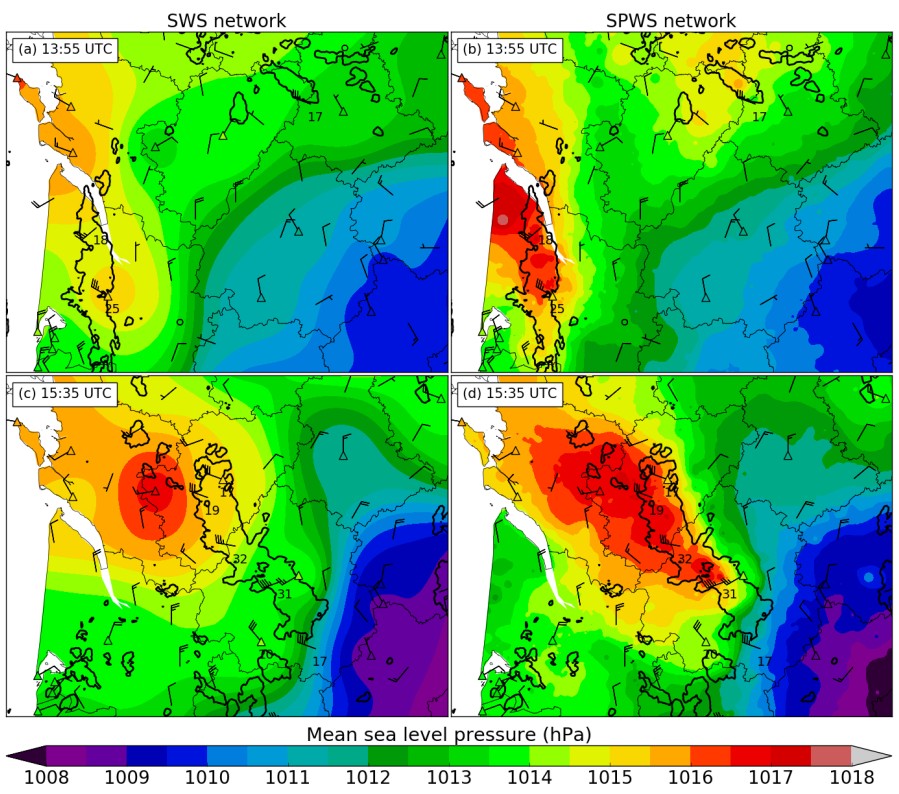

**Figure 12.** As in Fig. 11, for the 4 July case. MSLP (a,c) SWS analyses and (b,d) SPWS analyses at (a,b) 13:55 UTC and (c,d) 15:35 UTC 4 July.

### 6.1.3   28 August 2018

At 19:05 UTC 28 August 2018, a squall line was moving north-northeastwards. Radar indicated a broad area of strong reflectivities over 40 dBZ. At the surface, high wind gusts were measured: $27 \mathrm{~m\,s^{-1}}$ near the center of the line at 18:48 UTC, $31 \mathrm{~m\,s^{-1}}$ in its southern part at 18:58 UTC and $28 \mathrm{~m\,s^{-1}}$ in the northern part of the line at 19:04 UTC. These strong gusts

5 were associated with jumps in MSLP: for example, in the northern SWS, the wind gust was preceded by a 2.8 hPa surge in 4 min between 19:00 UTC and 19:04 UTC. These observed strong MSLP gradients at the gust front are in agreement with the sharper gradients shown by the SPWS analysis in Fig. 13b compared to the SWS analysis in Fig. 13a. Also, the location of the gust front, especially in its eastern part is different between analyses: the SPWS analysis in Fig. 13b shows that the front has not reached two SWSs located south-east of the "32" number whereas according to the SWS analysis in Fig. 13a, they were

10 already concerned by the front. SWS observations show weak gust speed, supporting the accuracy of the SPWS analysis.

At 20:35 UTC, SWSs observed strong gusts at the rear of the pressure front extending from north-northwest to south-east (Fig. 13c). MSLP field of the SWS analysis do not explain such gusts, especially in the northern parts of the figure. The SPWS analysis in Fig. 13d reveals a MSLP surge under the northern convective line, which is missed by the SWS analysis. This high
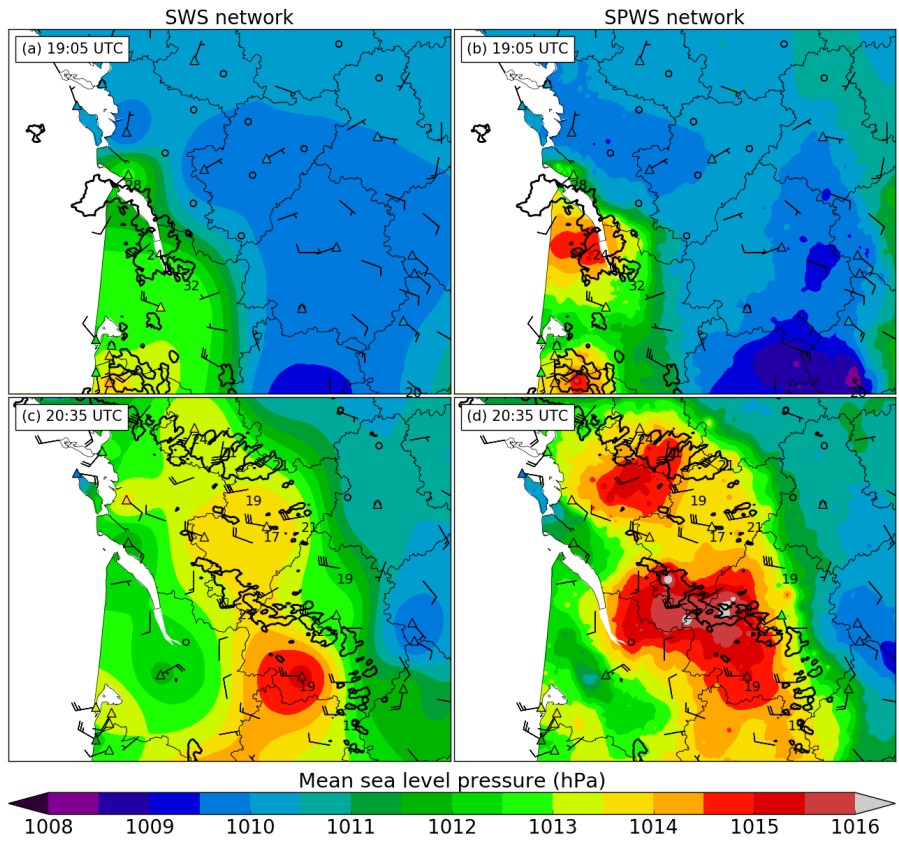

**Figure 13.** As in Fig. 11, for the 28 August case. MSLP (a,c) SWS analyses and (b,d) SPWS analyses at (a,b) 19:05 UTC and (c,d) 20:05 UTC 28 August.

seems to cause the wind gusts between 19 and 23 m s$^{-1}$ observed near the surface. The SWS analysis also indicates lower surge in pressure at the south, compared to SPWS analysis, probably because no SWS is located at its center. The SPWS analysis provides insight about the convection organization, because it shows that the convective line south and the one north are two independent squall lines: two pressure surges with separated wake lows at the rear are visible in the SPWS analysis, each corresponding to the theoretical structure of a squall line (Johnson and Hamilton, 1988). Even if their gust fronts merged in some areas, which triggered the formation of scattered cells between them, the pressure field and the direction of the main cells indicate that these two squall lines are not merging. The independence of the two lines is unclear with the SWS analysis in MSLP or with radar reflectivities. The independence of these two squall lines seems to be accurate because afterwards, according to radar reflectivities, the convective cells associated with the two MCS followed slightly different directions.




## 6.2 Contribution of PWSs to temperature and humidity analyses

The contribution of PWSs in two situations is shown. Measurements of surface pressure, temperature and relative humidity of SPWS network allow to compute derived quantities such as virtual temperature $T_v$, temperature of a dry air parcel which has the same density as the humid air considered, and the virtual potential temperature associated $\theta_v$, which is related to buoyancy and is identified as pertinent to track cold pools by, e.g., Drager and Van den Heever (2017). $\theta_v$ is computed as follows:

$$\theta_v = T_v \left( \frac{P_0}{P} \right)^{\frac{R_0}{M c_p}} \tag{11}$$

with $P$,$M$,$R_0$ defined in Eq. (1), $T_v$ defined in Eq. (3), $P_0 = 1000$ hPa the standard reference pressure and $c_p = \frac{7}{2} \frac{R_0}{M}$ is the specific heat capacity at a constant pressure.

### 6.2.1 4 July 2018

In the morning of 4 July 2018, before the line studied in Sect. 6.1.2 affected the Bordeaux region, isolated storms formed over the south-west of France and moved north-northeastwards. At 12:55 UTC, a cluster of convective cells was seen by radar (Figs. 14a,b). The storm identified as a supercell, south-east of this cluster, produced tennis-ball sized hail few minutes later. With the SPWS network, a temperature drop of about 6 °C in 10 min under this cell and a rapid rise in relative humidity are observed. It is confirmed by a SWS which was on the path of this convective cell few minutes later: temperature dropped from 23.9 to 17.4 °C and relative humidity rose from 67 to 94 % in 30 min. In the upper-right part of the Fig. 14b, a storm crossed the eastern part of the city of Limoges, where the spatial density of PWSs is high. The path of the cell is visible in temperature and relative humidity with more details with SPWSs than with SWSs only. The SWS located west of Limoges measured a 71 % relative humidity while several PWSs, located closer to the path of the cell exceeded 80 %, values coherent with road station measurements of about 85 to 95 %. Also near the mouth of the Garonne, at the left of Fig. 14d, PWSs indicate high relative humidity between 70 and 90 % that disagrees with a SWS measured value of 60 %. The analysis of radar reflectivities indicates that two storms concerned this area, but the SWS was not directly reached by one of them. In this case, the higher density of PWSs gives fine details of features in temperature and humidity associated with deep convection.

Later this day, isolated storms also formed before the arrival of a squall line over the west of the Massif Central mountains. One in particular created a powerful cold pool: temperature dropped by 15.1 °C and relative humidity increased by 61 % in 1h50 at a SWS located near the center of it. At 15:35 UTC, additional weather stations not used to build analyses of Figs. 15a,b, measuring only temperature with a 6-min time step (measure at 15:36 UTC) were added on the figures to assess the quality of the temperature fields. Main differences in temperature are observed in the warm area between the cold pool and the west of the figures. Two additional weather stations agree with the increase in temperature proposed in this area by the SPWS analysis in Fig. 15b, especially north-west of the cold pool. A little decrease in temperature is also shown by the SPWS analysis south-west of the figure associated with a small convective cell but is not shown by the SWS analysis. In relative humidity, differences are also visible between Fig. 15c and Fig. 15d. In four areas, east, south-southeast, south-west and west-southwest of the


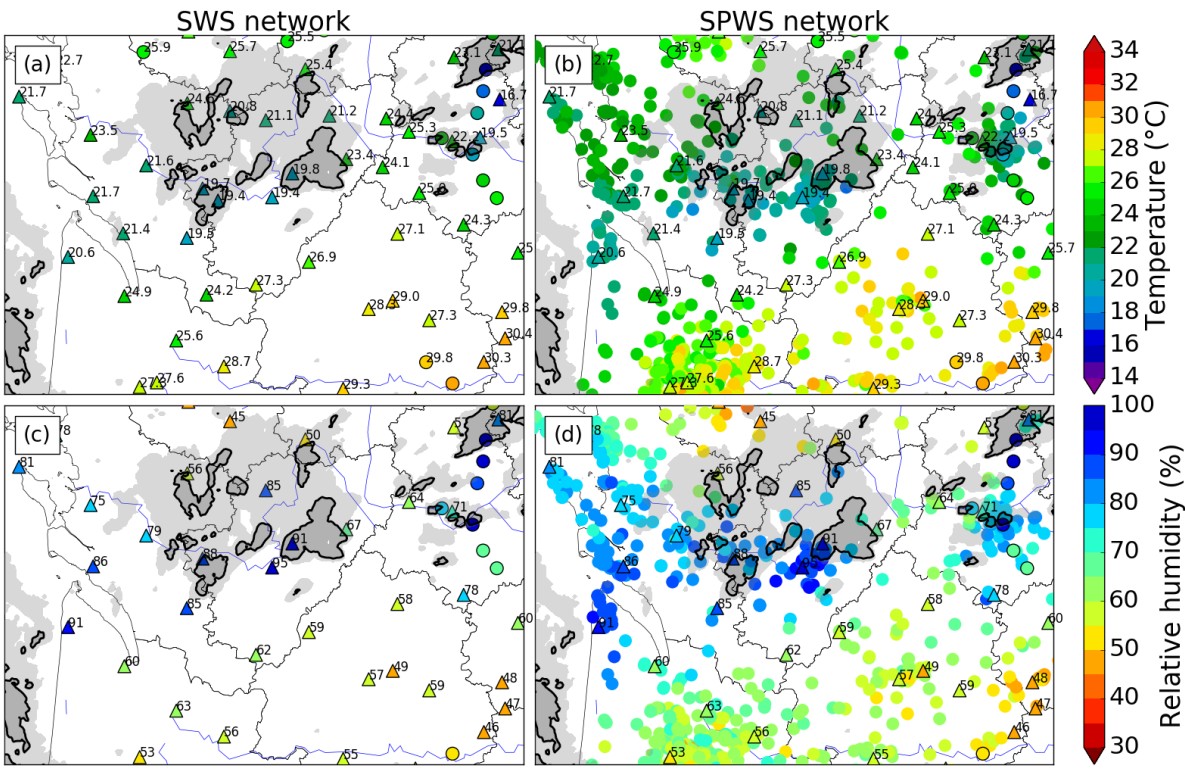

**Figure 14.** Observations of (a,b) temperature and (c,d) relative humidity at 12:55 UTC 4 July with (a,c) SWS network and (b,d) SPWS network. SWS measurements are shown by coloured triangles with black contours. Additional 6-min time steps stations are circled in black with values and road weather stations are only circled in black. Reflectivities >18 dBZ: light grey, >40 dBZ: grey with black contours.

central cold pool, relative humidity is higher in SPWS analysis than in SWS analysis. After looking at the reflectivity field, the observed increases are temporally and spatially consistent with the passage of isolated convective cells over these four areas.

In this case, the development of a cold pool that extended over time in all directions is observed. The extension of the cold pool towards south, east and west initiated deep convection in these directions between 15:35 UTC and 15:55 UTC (Fig. 16).

5    South of the cold pool, small cells advected in a south-westerly flux brutally strengthened near the cold pool boundary. Two PWSs located south informed about the cold pool propagation speed and observed its extension southwards. West of this isolated cold pool, secondary convective initiation was observed before the MCS cold pool located left of the figure and the isolated cold pool merged. Areas where convective storms were triggered have higher $\theta_v$ in SPWS analysis (Fig. 16f) than in SWS analysis (Fig. 16e). At 16:15 UTC, the main difference between both analyses concerns a warm zone between the MCS

10   cold pool at the west and the isolated cold pool. Warm conditions with temperatures around 24 °C are observed by SPWS network while SWS network indicates temperatures between 20 °C and 22 °C. These higher temperatures are confirmed by two additional weather stations indicating temperatures 2 °C to 4 °C higher than the temperature given by SWS analysis.


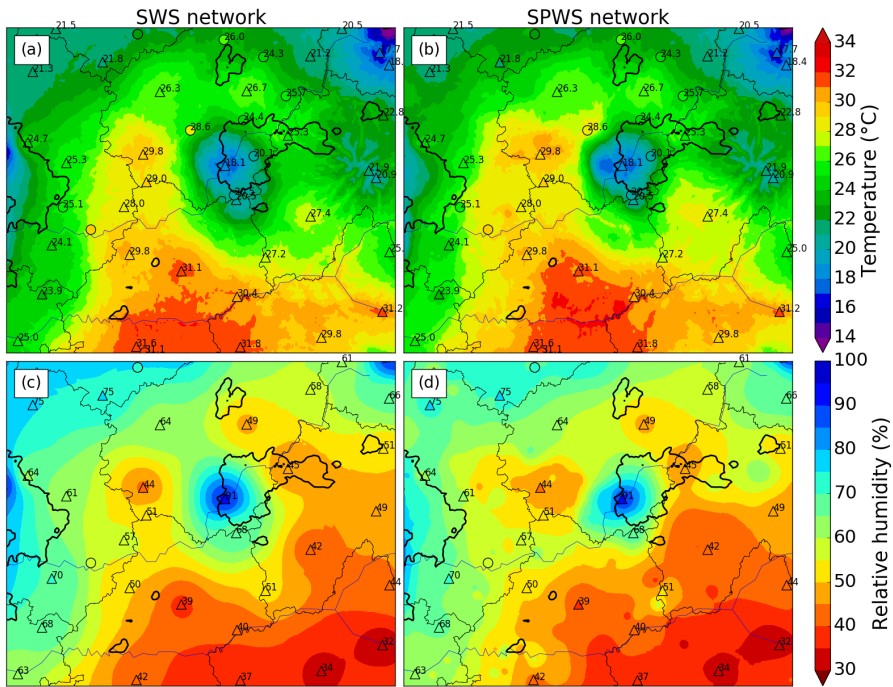

**Figure 15.** (a,b) Temperature and (c,d) relative humidity (a,c) SWS analyses and (b,d) SPWS analyses at 15:35 UTC 4 July. Station measurements are as in Fig. 14. Reflectivities over 40 dBZ are materialized by bold black contours.

These differences in temperature mainly explain the differences in potential virtual temperatures observed in this area between Fig. 16e and Fig. 16f: $\theta_v$ between 26 °C and 30 °C are indicated by SPWS analysis against 24 to 26 °C in SWS analysis.

### 6.2.2 15 July 2018

On 15 July 2018, an isolated thunderstorm formed at the south-east of the Arcachon Bay (Fig. 17), where converging winds
5  due to sea breeze were observed. A SWS was located near the initiation point and measured warm temperatures around 32 °C
before the initiation and relative humidity around 48 %. North of this station, other inland SWSs with comparable temperatures
measured lower relative humidity between 34 % to 38 % at the same time. Steep gradients of 2-m temperature and 2-m relative
humidity were observed with 5 °C temperature change within 40 km and 22 % relative humidity change within 30 km. The
thunderstorm moved north-eastwards and triggered the initiation of few convective towers. The cluster of convective towers
10  then split in two main cells, one headed east and the other north (Fig. 17). At the surface, both cells induced drops in temperature
and rises in relative humidity. However, at 14:45 UTC, no SWS was directly under the path of the cells: only little decrease
in temperature and little increase in relative humidity were observed at long range. At the same time, several PWSs recorded
drops in temperature and strong increases in relative humidity associated to radar reflectivities above 40 dBZ. At the next
time steps, several SWSs detected features of similar or higher amplitude that supported the consistency of PWS observations.


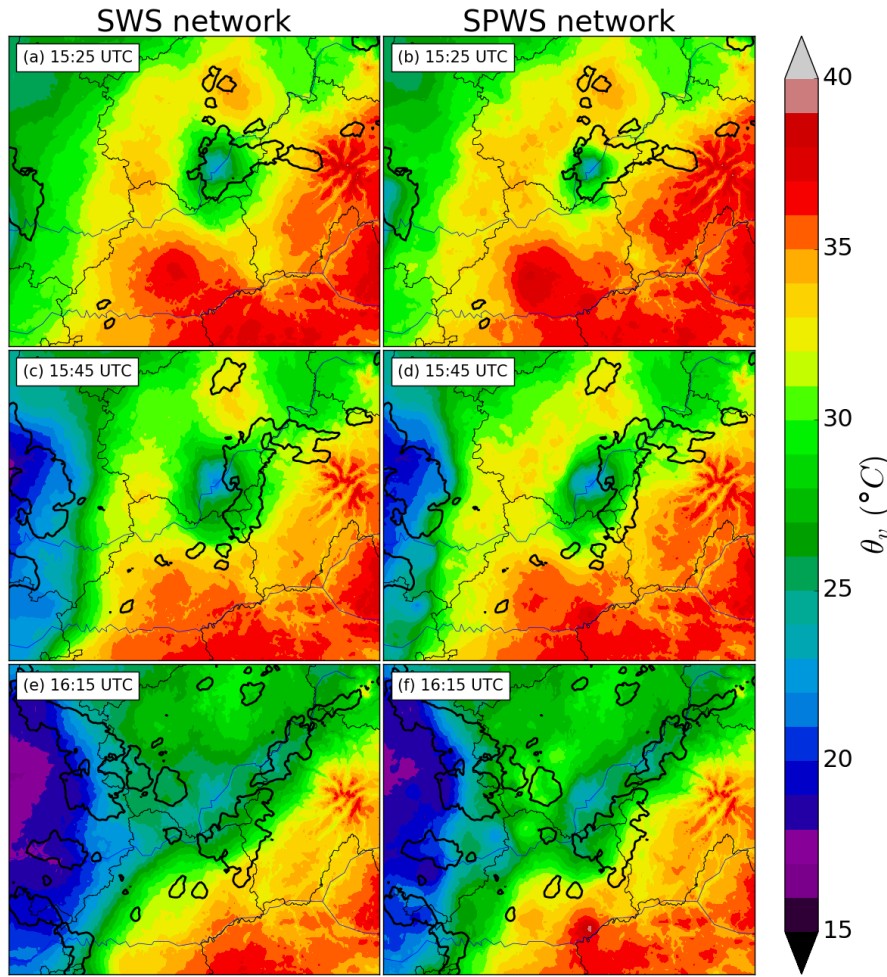

**Figure 16.** $\theta_v$ (a,c,e) SWS analyses and (b,d,f) SPWS analyses at (a,b) 15:25 UTC, (c,d) 15:45 UTC and (e,f) 16:15 UTC 4 July. Reflectivities over 40 dBZ are materialized by bold black contours.

In this case, adding PWSs gives insight into the extension of the cold pool associated with thunderstorms and confirms that precipitation are reaching the surface. It leads to differences between SWS and SPWS analyses up to 6 °C in temperature, 30 % in relative humidity and 8 °C in virtual potential temperature in areas concerned by thunderstorms. The increased spatial density contributes to a finer mapping of areas that were cooled, or, on the contrary, areas where convective cells haven't cooled
5   the atmosphere near the ground, which may be the location of further convective initiations.

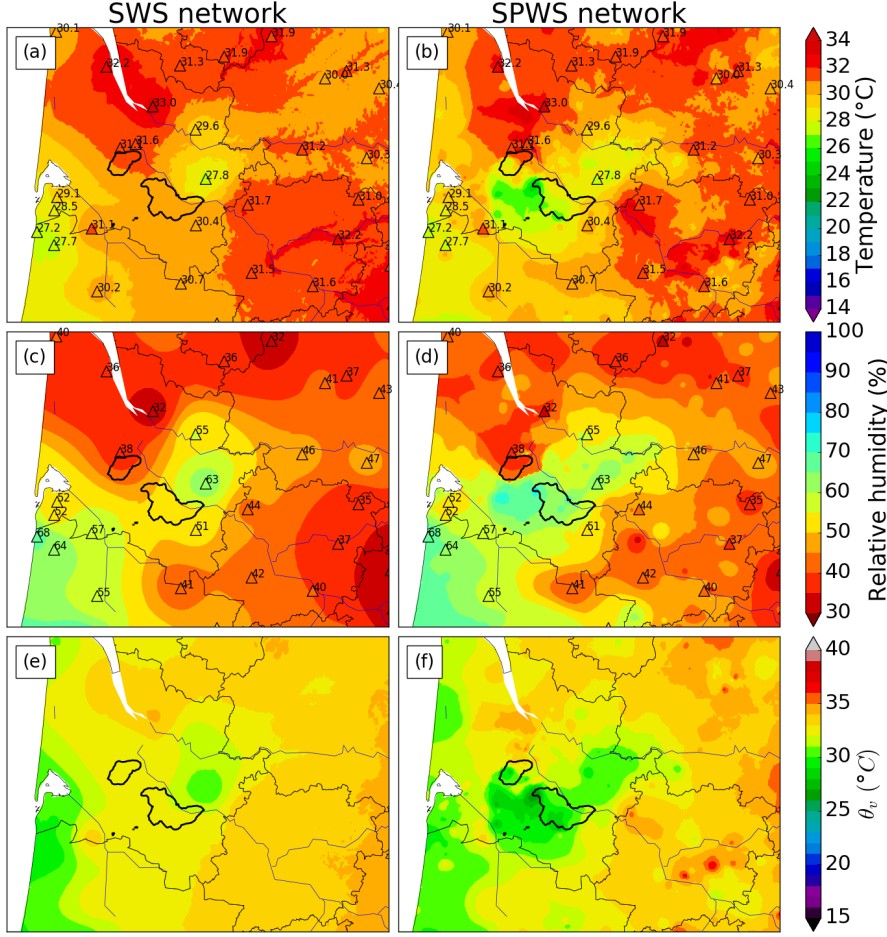

**Figure 17.** (a,b) Temperature, (c,d) relative humidity and (e,f) virtual potential temperature (a,c,e) SWS analyses and (b,d,f) SPWS analyses at 14:45 UTC 15 July. Station measurements are as in Fig. 14. Reflectivities over 40 dBZ are materialized by bold black contours.

## 7  Conclusions

Some PWS networks now sample the atmosphere at high spatial and temporal resolution: the Netatmo network, on which this study focused, constitutes a network of weather stations of identical sensors with unprecedented density available in near real-time, with a minimum 5-min temporal resolution.

5      Adding raw PWS data in observed surface analyses strongly deteriorates the RMSE calculated by LOOCV in comparison with using only SWS analyses. It increased the $\mathrm{RMSE_{LOOCV}}$ from 41 to 72 % in temperature and from 6 to 31 % in relative humidity depending on the case, showing the negative contribution of PWSs if they are not properly preprocessed and quality-controlled.


An automatic processing including a quality control was designed and based on comparison with SWS analyses, during short temporal windows. Median systematic errors are computed and corrected at first for all parameters. Simple quality checks with four steps in pressure, and only three in temperature and humidity were designed. In temperature and humidity, the main step eliminates PWS time series with too high departures compared to SWS analyses in RMSE. The RMSE threshold is

automatically chosen by minimizing a $\mathrm{RMSE_{LOOCV}}$ taking the SWS network as the validation dataset. In pressure, an algorithm performing at each step a LOOCV taking all stations as the validation dataset was developed. The PWS providing the highest $\mathrm{RMSE_{LOOCV,k}}$ is eliminated at each step. The algorithm stops when the first local minimum in $\mathrm{RMSE_{LOOCV}}$ is obtained. Over the four case studies, the mean number of PWS observations kept after processing is $91~\% \pm 3~\%$ in MSLP, $37~\% \pm 7~\%$ in temperature and $39~\% \pm 12~\%$ in relative humidity. On average, the number of available observations is multiplied by 134 in

MSLP, by 11 in temperature and by 14 in relative humidity.

A LOOCV was performed in several convective cases to validate the method on the SWS observations. Results over Metropolitan France show a substantial decrease of the $\mathrm{RMSE_{LOOCV}}$ between 73 and 77 % in MSLP. Decreases in $\mathrm{RMSE_{LOOCV}}$ are also observed in temperature between 12 and 23 % while the decrease in relative humidity reaches 17 to 21 %. These scores quantitatively show that adding PWSs to SWSs improves the accuracy of surface analyses, especially in MSLP.

Qualitatively, fine-scale structures partly or not seen by SWS network only showed up in MSLP, temperature and humidity when PWS and SWS networks were combined in several case studies. In MSLP, pressure surges accompanying squall lines were observed as well as wake lows at the rear of these lines. Pressure surges accompanying individual cells were also observed. A crescent-shaped MSLP structure was observed approximately one hour prior to the transition of a squall line in a bow echo. Also, a gust front still producing wind gusts up to $25~\mathrm{m\,s^{-1}}$ was detected and its movement tracked while its associated

convective system exhibited rapid decay in radar reflectivities. These structures were consistent with the movement of storm systems detected by radar and with observed variations of MSLP or wind speed at SWSs locations. All these structures observed with the SPWS network were only partly or not visible at all with SWS observations only.

In temperature and humidity, temperature drops and humidity surges accompanying most of the cells were observed, giving a storm signature at the ground in temperature and humidity. The virtual potential temperature $\theta_v$ derived from surface ob-

servations was spatialized at an unprecedented spatial resolution thanks to PWS contributions. In two case studies, cold pool propagation and secondary convective initiation over areas of high virtual potential temperatures, i.e. favorable locations for near surface parcel lifting were observed. Future work will focus on using these observations for the validation of fine-scale numerical simulations of convective cases. The goal is to figure out whether these simulations reproduce all the phenomena observed by these PWSs, and investigate the potential differences as a preparatory work before a possible assimilation of

these new data. Future work may also include the development of an operational tool to display these PWS measurements, especially to track convective structures, at a $5\text{-}\mathrm{min}$ temporal resolution if possible. Points as early discrimination between surface-based and elevated convection, as well as favoured locations for convective initiation or secondary cell development, already highlighted by Clark et al. (2018) may also be investigated.





*Author contributions.* This work was carried out by MM as part of his PhD thesis under the supervision of OC. MM and OC collaborated, interpreted the results and wrote the paper.

*Competing interests.* The authors declare they have no conflict of interest.

*Acknowledgements.* We wish to thank the Météo-France Direction des Systèmes d'Observations for the field tests of the PWSs they per-
5   formed and for all datasets provided. We acknowledge the use of imagery from the NASA Worldview application (https://worldview.
earthdata.nasa.gov), part of the NASA Earth Observing System Data and Information System (EOSDIS).



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
