# Peer review of "Contribution of personal weather stations to the observation of deep-convection features near the ground"

_Natural Hazards and Earth System Sciences, 2019_

## Referee Comment (RC1) · Anonymous Referee #1 · 9 Aug 2019

Review of "Contribution of personal weather stations to the observation of deep-convection features near the ground" by Marc Mandement and Olivier Caumont (nhess-2019-229), submitted to Natural Hazards and Earth System Sciences (NHESS).

**Summary:**

The paper describes four big-impact weather events in detail, where the authors show the added benefit of using temperature, pressure and relative humidity observations

by Netatmo personal weather stations in addition to the standard weather station networks. The increased detail that can be obtained is demonstrated both quantitatively and qualitatively, provided the PWS observations are properly quality controlled, for which a successful method is proposed. This is a very important topic, as high resolution weather monitoring is useful both for (urban) operational applications and for validation of numerical models, and this data source shows a lot of potential.

The article is well written with figures of high quality. Both the structure of the paper and the figures/schematics are helpful in the clarity of the overall manuscript. The message that PWS observations improve the resolution at which weather events can be described is founded in a complete and well-reasoned analysis. I recommend this manuscript to be published after minor revisions, taking into consideration the following comments.

**Major comments:**

- Section 4.1: the method relies considerably on the notion that the reported elevation of the PWSs are accurate. Do the authors have any idea how valid the assumption is that the reported PWS altitudes are correct?

- Section 4.4: the quality control seems to be based on checks where time series of a PWS are compared with those of SWS. This seems to be based on time series over the complete event and can therefore only be performed afterwards. The authors could consider a variation of the method that could be applied in real-time, or comment on the possibility of operational implementation in the discussion.

**Minor comments:**

- P2, line 8: comma after 'evolution'.

- P2, line 13: consider using 'Additionally,' instead of 'Also'.

- P2, line 19-20: should there indeed be two citations?

- P2, line 23: 'leaded' should be 'led'.

- P2, line 26: 'establish' should be 'establishing'.

- P2, line 31: de Vos et al. (2017) uses precipitation data retrieved from the Wundermap platform, not the Netatmo platform, and only part of PWSs are of type Netatmo. The paper does discuss Netatmo precipitation measurements in particular in an experimental context evaluating 3 Netatmo gauges. This sentence may be adjusted to reflect that. You could consider referencing de Vos et al. (2019) doi:10.1029/2019GL083731, which focusses on precipitation measurements from Netatmo PWSs only.

- P3, table 1: consider replacing 'Hour of...' for 'start time' and 'end time', as the periods don't start or end at a rounded clock hour.

- P3, line 4: consider leaving out ', focussing'.

- P3, line 10-14: the section references are incorrect.

- P3, line 24: 'the 26th'

- P4, line 1: no 'the' before 25.

- P4, line 3: 'However' instead of 'But'.

- Section 2.1 – 2.4: very well explained and informative. However, no source is provided for the number of fatalities, rescue operations, etc. This (likely a news archive?) may be mentioned once in the beginning of the section.

- P10, line 11: 'Netatmo provided in near real time only 10 min time step measurement', how does this follow from the previous statement that the mean time step is 5 min?

- P11, line 4: 'linear interpolation': does this mean that a measurement is attributed to the 10 min time stamp it is closest to in time? Or is a weighted average constructed? How are data gaps handled in that case? Some more information would be desirable.

- P11, line 11: Some more background on the field tests, i.e. the duration, would be desirable. It may also be constructive to mention some quantitative errors found in other papers for comparison, see e.g. Fig 2 in Meier et al. (2017) doi:10.1016/j.uclim.2017.01.006.

- P12, Figure 7: the figure is very helpful in clarifying the method. I would prefer to replace 'x' by a dot, as it reads as a letter instead of a multiplication sign.

- P12, line 4: 'The radius is larger for pressure because it is the minimum radius allowing to cover the entire Metropolitan France' is this because of the low number of SWSs that measure pressure, and are therefore the ranges of SWSs and PWSs the same? Is there a different expectation in spatial variability for each of the variables, and if so, is that also a factor to consider when determining the range?

- P13, line 25: the citation should be '(de Ruiter, 2016)' as Tom is a first name.

- P14, line 8: do the authors mean that altitude varies with spatial distance and therefore values grid points at a some distance may differ from the value at the station?

- P14, line 28: 'less than half of the measurements are available': when are measurements not available? Given the linear interpolation of time lines, how large

should a data gap be to result in an unavailable 10 min interval? (This can be explained on P11, not necessarily here.)

- P15, line 22: 'tends to' should be replaced by 'approaches'.

- P18, line 11: is the reference to Sect. 6 correct?

- P18, line 28: for clarity the authors may choose to change '73

- P18, line 30: 'on average' instead of 'in mean'.

- P19, line 3: 'the increase' instead of 'increase'

- P19, line 12: 'in mean' should be replaced by 'on average'.

- P20, line 9: for clarity the authors may choose to change '17

- P20, line 11: 'in mean' should be replaced by 'on average'.

- P21, Figure 10: consider increasing the symbol size in subfigure (a) for improved readability.

- P27, line 5-6: 'Two PWSs ... extension southwards.' Sentence is unclear, please rephrase.

- P33, line 13: citation lacks 'W.', see the 2018 citation below.

---

## Referee Comment (RC2) · Anonymous Referee #2 · 24 Aug 2019

The paper concerns the assessment of the added value of personal weather stations data, provided by Netatmo sensors, for the assessment of the near surface spatio-temporal evolution of deep moist convective phenomane.

The paper is very timely from a research standpoint, well-written and based on a convincing methodological approach.

Only some major comments are raised hereafter:

1) page 11, line 31: "The gridding method used is the inverse distance weighting (IDW)". Question: are the results sensitive to the adopted gridding method?

[Figure]

2) page 12, lines 1-5: how the limit distances between grid points and station location were selected?

3) the quality control is based on complete timeseries and it seems to be performed only in post-event phase. Can the authors consider the possibility to apply a quality flag to a certain station based on several post-event timeseries validation? this probably would pave the way also to a real time usage of these stations.

---

## Author Comment (AC2) · 15 Nov 2019

The authors thank the referee #2 for the thoughtful and constructive comments. Our response to comments is in the attached pdf supplement which also contains the revised manuscript with changes tracking.

Please also note the supplement to this comment:
https://www.nat-hazards-earth-syst-sci-discuss.net/nhess-2019-229/nhess-2019-229-AC2-supplement.pdf
2019-229, 2019.

---

## Author Response (AR1)

**Reply to referees - NHESS-2019-229**

We thank the referees for their thoughtful comments, which we have addressed below. Comments from referees are in *italics*, indicated by  $\bullet$ ; our response is in plain text, indicated by >. Changes in manuscript are indicated by  $\mathbb{Z}_{2}$  under our response: unchanged parts of the manuscript are in plain text, deleted parts are in red-and added parts are in blue.

**5 Reply to anonymous referee #1**

**Major comments:**

20

25

• Section 4.1: the method relies considerably on the notion that the reported elevation of the PWSs are accurate. Do the authors have any idea how valid the assumption is that the reported PWS altitudes are correct?

> The paragraph was probably unclear: the gridding method of the section 4.1 does not use at all PWSs elevations, even when
 SWS and PWS are gridded together. For temperature and surface pressure which vary strongly with altitude, the linear regression only uses SWS observations and so SWS elevations which are known with accuracy. In this whole section, only latitude and longitude of PWSs are used, during the IDW calculation. The paragraph was modified:

For temperature and surface pressure, because they vary strongly with altitude, a linear regression of the SWS observations used in the gridding with respect to the altitude is performed first. After that, the residuals (i.e. the difference between the values obtained by linear regression and the observations) are gridded as shown in Fig. 7a method [2], and then added to the grid derived from the linear regression. The linear regression uses only the SWS observations used in the gridding.

> To answer the question how accurate PWS altitudes are, in section 4.2, the PWS altitudes given by Netatmo users are compared to a digital elevation model. For the 26 May case for example, the quantiles 1, 5, 25, 50, 75, 95, 99 of the departures between PWSs altitudes and the DEM are -143 m, -19 m, -3 m, 0 m, 2 m, 24 m, 83 m. PWS altitudes include obvious outliers which are outside the range of Earth's terrain altitudes. For this case, 88% of the PWSs providing altitudes have departures in absolute value of less than 15 m.

• Section 4.4: the quality control seems to be based on checks where time series of a PWS are compared with those of SWS. This seems to be based on time series over the complete event and can therefore only be performed afterwards. The authors could consider a variation of the method that could be applied in real-time, or comment on the possibility of operational implementation in the discussion.

> The following answer was added at the end of the Sect. 4.4:

In real-time, we do not have access to the complete time series. A variation of the method that could be applied in real-time is using time series over a rolling period of the 24 last hours ending at the time of the analysis instead of the time series over

- 30 a complete event. Then, every 10 min the automatic processing would be launched for each analysis produced. This method implies that the algorithm runs in less than 10 min, which is not the case for the current algorithm. It takes around 1 h to perform the quality control over 24 h of measurements on a computer with a central processing unit (CPU) with 4 cores and 16 Go of random-access memory. To increase the processing speed, one or several available computing nodes with 24 cores each could be used because the algorithm is partially parallel. Parts of the algorithm that are still sequential could be
- 35 parallelized. In addition, the LOOCVs in the quality control could be modified because there are the most time-consuming parts of the algorithm. For temperature and humidity LOOCVs provide thresholds and for pressure the LOOCV eliminates a small number of PWSs one by one. Thus, the algorithm at a given time can use the temperature and humidity thresholds as well as the list of PWSs eliminated that were computed by the algorithm launched one hour before.

**Minor comments:**

40 • P2, line 8: comma after 'evolution'.

> Corrected: ⊯ evolution, still

• P2, line 13: consider using 'Additionally,' instead of 'Also'.

> Corrected:

∠ Also-Additionally, there 5

• P2, line 19-20: should there indeed be two citations?

> Agreed, the citation was divided in two:

Z Sobash and Stensrud (2015); Gasperoni et al. (2018) showed It was shown that 5-min assimilation of mesonet data (Sobash and Stensrud, 2015) or nonconventional data (Gasperoni et al., 2018), mostly thermodynamic observations, improved forecasts of convection initiation.

10

• P2. line 23: 'leaded' should be 'led'.

> Corrected: measurements leaded-led to

• P2, line 26: 'establish' should be 'establishing'.

> Corrected: 15

∠ and establish establishing robust

• P2, line 31: de Vos et al. (2017) uses precipitation data retrieved from the Wundermap platform, not the Netatmo platform, and only part of PWSs are of type Netatmo. The paper does discuss Netatmo precipitation measurements in particular in an experimental context evaluating 3 Netatmo gauges. This sentence may be adjusted to reflect that. You could consider referencing de Vos et al.(2019) doi:10.1029/2019GL083731, which focuses on precipitation measurements from Netatmo PWSs only.

20

30

> The referee is right, the paragraph was modified:

🖉 For precipitation, de Vos et al. (2017, 2018) showed that rain gauges produced by Netatmo, a PWS manufacturer, de Vos et al. (2017) showed that dense PWS networks can be used for urban rainfall monitoring , capturing in Amsterdam and 25 are able to capture well small-scale rainfall dynamics in Amsterdamaccording to a simulation study under ideal measurements conditions (de Vos et al., 2018). Subsequently, a real-time quality control algorithm of rain gauges produced by Netatmo, a PWS manufacturer, was developed in order to complement traditional networks for operational rainfall monitoring (de Vos et al., 2019).

• P3, table 1: consider replacing 'Hour of...' for 'start time' and 'end time', as the periods don't start or end at a rounded clock hour.

> Corrected:

⊯ Hour of beginning Start time (UTC) Hour of end End time (UTC)

• P3, line 4: consider leaving out ', focusing'.

**35 > Corrected:**

⊯ at midlatitudes <del>, focusing</del> over France

• P3, line 10-14: the section references are incorrect.

> Corrected:

🖉 First, in Sect. 12 this study describes interesting convective cases of the spring and summer 2018 over France. In Sect. 23, a presentation of the different weather station networks used in the study is made. The processing including a quality control of PWSs measurements is detailed in Sect. 34, followed by the validation performed against SWSs in Sect. 45. Then, a focus on

5 some features observed during the different convective cases is made in Sect. 56 to evidence the positive contribution of PWSs.

• P3. line 24: 'the 26th'

> Corrected:

• P4. line 1: no 'the' before 25.

> Corrected: 10 ∠ between the 25 May 22:00 UTC

• P4. line 3: 'However' instead of 'But'.

> Corrected: But However it recorded

15 • Section 2.1 - 2.4: very well explained and informative. However, no source is provided for the number of fatalities, rescue operations, etc. This (likely a news archive?) may be mentioned once in the beginning of the section.

> A paragraph was added in the beginning of the Sect. 2:

isted in each case come from internal reports of the French emergency management agency (Sécurité Civile), press releases from the prefectures, press releases from the French distribution grid operator (Enedis) and press archives (France 3, Sud-Ouest).

20

30

• P10, line 11: 'Netatmo provided in near real time only 10 min time step measurement', how does this follow from the previous statement that the mean time step is 5 min?

> The paragraph was modified to explain that Netatmo indicates a mean time step of 5 min and provides only 10 min time step measurements:

the mean time step between two measurements indicated by the manufacturer is 5 min <del>but;</del> it may sometimes vary because 25 PWS owners can also perform additional on-demand measurements. However, Netatmo provided in near real time only 10 min time step measurements, which is the minimum time step used in this study.

• P11, line 4: 'linear interpolation': does this mean that a measurement is attributed to the 10 min time stamp it is closest to in time? Or is a weighted average constructed? How are data gaps handled in that case? Some more information would be desirable.

> The interpolation is indeed linear and not a nearest-neighbour interpolation, meaning that the result of the interpolation is the raw measurement itself if both interpolation and measurement times are simultaneous, otherwise the result of the interpolation is a weighted average of the two closest raw measurements around it if they are spaced by less than 700 s. In all other cases, the result of the interpolation is a missing value. The following sentence was added to explain how data gaps are handled:

ightharpoint for the two closest measurements around the interpolation time are separated by a period of 35 700 s or more.

• P11, line 11: Some more background on the field tests, i.e. the duration, would be desirable. It may also be constructive to mention some quantitative errors found in other papers for comparison, see e.g. Fig 2 in Meier et al. (2017)doi:10.1016/j.uclim.2017.01.006.

> The paragraph was modified:

I field tests were realized at Météo-France with during 80 days by comparing 3 Netatmo PWSs to a reference SWS including a platinum temperature sensor with an accuracy of  $\pm$  0.23 °C between -20 and 40 °C, and a Vaisala HMP 110 humidity sensor with an accuracy of  $\pm$  2.5 % between 0 and 40 °C. Two white plastic radiation shields naturally ventilated are used:

- 5 the reference sensors were in a Socrima BM0 1195 model while the Netatmo outdoor modules were in a larger Socrima BM0 1161. Tests show errors in temperature of about 0 °C  $\pm$  0.9 °C in median and 95 % interval compared to a reference SWS, and errors in relative humidity of about +3 %  $\pm$  7 % in median and 95 % interval. These tests have been performed with a supplementary radiation shield: they show a correct quality of temperature and humidity sensors when properly protected but do not give insights about their accuracy without this shield. the radiation shield. They show the same diurnal cycle of
- 10  $\Delta T_{\text{Netatmo-SWS}}$  as the Fig. 2 of *Meier et al.* (2017) but with a lower amplitude: the median remains in the range 0 °C ± 0.5 °C for all hours of the day.

• P12, Figure 7: the figure is very helpful in clarifying the method. I would prefer to replace 'x' by a dot, as it reads as a letter instead of a multiplication sign.

> The 'x' was replaced by a dot.

• P12, line 4: 'The radius is larger for pressure because it is the minimum radius allowing to cover the entire Metropolitan France' is this because of the low number of SWSs that measure pressure, and are therefore the ranges of SWSs and PWSs the same? Is there a different expectation in spatial variability for each of the variables, and if so, is that also a factor to consider when determining the range?

> The fact that the radius is larger is due to the small number of SWS pressure sensors but also because stations with altitude higher than 750 m are discarded. There is no preconceived expectation in spatial variability for each of the variables. Every maximum range is a trade-off between the smallest possible distance that limits the extrapolation of small scale features and between a larger distance keeping enough stations to limit the influence of a single station over the surrounding grid points. See also the second major comment of referee #2: some sentences were added in Sect. 4.1.

• P13, line 25: the citation should be '(de Ruijter, 2016)' as Tom is a first name.

25 > Corrected:

• P14, line 8: do the authors mean that altitude varies with spatial distance and therefore values grid points at a some distance may differ from the value at the station?

> Exactly, the paragraph was modified to be clearer:

- 30 ∠ For surface pressure and temperature, because they vary rapidly with altitude , PWS time series are not compared directly to the SWS analyses at the closest grid point. Indeed, the altitude of the which itself varies rapidly with spatial distance in mountainous regions, the value at the PWS closest grid point may be really different of the PWS actual one actual PWS value. That is why a-PWS time series are not compared directly to the SWS analyses at the closest grid point. A more precise calculation [...]
- P14, line 28: 'less than half of the measurements are available': when are measurements not available? Given the linear interpolation of time lines, how large should a data gap be to result in an unavailable 10 min interval? (This can be explained on P11, not necessarily here.)

> As recommended by the referee, it is explained in the sentence added for the P11, line 11 comment.

• P15, line 22: 'tends to' should be replaced by 'approaches'.

> Corrected:

 $\not$  For small values of x, p(x) tends to approaches m, the number of SWSs, and RMSELOOCV(p(x)) tends to approaches quite large values because of the small number of SWSs and their large spacing.

• P18, line 11: is the reference to Sect. 6 correct?

5 > Yes

• P18, line 28: for clarity the authors may choose to change '73

> The sentence was modified:

 $\measuredangle$  a decrease in RMSE reaching ranging from 73 % to 77 % is observed.

• P18, line 30: 'on average' instead of 'in mean'.

10 > Corrected:

✗ multiplied by 134 in mean on average over

• P19, line 3: 'the increase' instead of 'increase'

> Corrected:

 ∞ 1.45 °C and the increase in RMSE

15 • P19, line 12: 'in mean' should be replaced by 'on average'.

> Corrected:
 ∞ multiplied by 11 in mean on average over

• P20, line 9: for clarity the authors may choose to change '17

> The sentence was modified:

20 *\land* a decrease in RMSE reaching ranging from 17 % to 21 % is observed.

• P20, line 11: 'in mean' should be replaced by 'on average'.

• P21, Figure 10: consider increasing the symbol size in subfigure (a) for improved readability.

25 > Done, the size was increased.

• P27, line 5-6: 'Two PWSs ... extension southwards.' Sentence is unclear, please rephrase.

> The sentence was rewriten:

∠ South of the cold pool, small cells advected in a south-westerly flux brutally strengthened near the cold pool boundary. Two PWSs<del>located south informed about the cold pool propagation speed and observed its extension southwards</del>, located in

30 this area, observed the southern boundary of the cold pool: SPWS analyses between 15:25 UTC and 15:45 UTC show the  $\theta_v = 31$  °C limit extending southwards faster than in SWS analyses, near the location where convective cells strengthened. Fine-scale observations of cold pool boundaries may help to identify locations where lifting is favoured.

• P33, line 13: citation lacks 'W.', see the 2018 citation below.

> Corrected:

**Reply to anonymous referee #2**

Major comments:

• 1) page 11, line 31: "The gridding method used is the inverse distance weighting(IDW)". Question: are the results sensitive to the adopted gridding method?

5 > To study the sensitivity to the gridding method, we slightly modified the gridding method. The power factor of the IDW was set to one instead of two. For the 26 May case, results are shown in Table 1 for MSLP, in Table 2 for temperature and in Table 3 for relative humidity. The following paragraph was added in Sect. 5.3:

It is study the sensitivity to the gridding method, we slightly modified it for the 26 May case. The power factor of the IDW was set to one instead of two. We observe little sensitivity to the change of the power factor. With a power factor of one (respec-

10 tively two), for SPWS the MSLP RMSE equals 0.118 hPa (0.099 hPa), the temperature RMSE equals 0.877 °C (0.889 °C) and the relative humidity RMSE equals 5.480 % (5.375 %). Decreases in RMSE reach 70 % (75 %) in MSLP, 16 % (16 %) in temperature and 17 % (21 %) in relative humidity with SPWS compared to SWS.

> Geostatistical methods such as kriging or more complex gridding methods were not evaluated on purpose because we think
 15 it is out of the scope of this study. This is a promising future work to compare the results obtained with IDW with all efficient gridding techniques known in scientific literature.

**Table 1.** Statistics of the LOOCV performed on SWS and SPWS observations of MSLP and validated on SWS observations for different gridding methods the 26 May 2018. The evolution in % is relative to the RMSE of SWS observations.

| Case st          | udy                   | 26 May 2018 |                      |       |                      |  |  |  |
|------------------|-----------------------|-------------|----------------------|-------|----------------------|--|--|--|
| Experir          | nent                  | Power       | factor $= 2$         | Power | Power factor = $1$   |  |  |  |
| Netwo            | ork                   | SWS         | SPWS                 | SWS   | SPWS                 |  |  |  |
|                  | Median                | 0.012       | -0.001               | 0.019 | 0.006                |  |  |  |
| MSLP error (hPa) | RMS
% of evolution | 0.404       | 0.099
75 % | 0.398 | 0.118
70 % |  |  |  |

**Table 2.** Statistics of the LOOCV performed on SWS, SPWS\_raw and SPWS observations of temperature and validated on SWS observations for different gridding methods the 26 May 2018. The evolution in % is relative to the RMSE of SWS observations.

| Case        | e study            |            | 26 May 2018           |                       |       |                       |                       |  |  |  |  |
|-------------|--------------------|------------|-----------------------|-----------------------|-------|-----------------------|-----------------------|--|--|--|--|
| Expe        | Ро                 | wer factor | = 2                   | Power factor $= 1$    |       |                       |                       |  |  |  |  |
| Network     |                    | SWS        | SPWS
_raw          | SPWS                  | SWS   | SPWS
_raw          | SPWS                  |  |  |  |  |
| Temperature | Median             | 0.009      | 1.122                 | 0.005                 | 0.005 | 1.075                 | 0.001                 |  |  |  |  |
| error (°C)  | RMS % of evolution | 1.060      | 1.823
+72 % | 0.889
-16 % | 1.046 | 1.641
+57 % | 0.877
–16 % |  |  |  |  |

**Table 3.** Statistics of the LOOCV performed on SWS, SPWS\_raw and SPWS observations of relative humidity and validated on SWS observations for different gridding methods the 26 May 2018. The evolution in % is relative to the RMSE of SWS observations.

| Cas                   | e study               | 26 May 2018  |                        |                       |                    |                       |                       |  |  |  |
|-----------------------|-----------------------|--------------|------------------------|-----------------------|--------------------|-----------------------|-----------------------|--|--|--|
| Exp                   | eriment               | Po           | wer factor             | = 2                   | Power factor = $1$ |                       |                       |  |  |  |
| Ne                    | SWS                   | SPWS
_raw | SPWS                   | SWS                   | SPWS
_raw       | SPWS                  |                       |  |  |  |
| Relative              | Median                | -0.238       | -3.268                 | -0.170                | -0.234             | -3.176                | -0.173                |  |  |  |
| humidity
error (%) | RMS
% of evolution | 6.820        | 8.898
+ 31 % | 5.375
-21 % | 6.616              | 8.264
+25 % | 5.480
-17 % |  |  |  |

• 2) page 12, lines 1-5: how the limit distances between grid points and station location were selected?

> Some sentences were added in Sect. 4.1 to explain our selection:

∠ Every maximum range is a trade-off between the smallest possible range that limits the extrapolation of small-scale features and a larger range keeping enough stations to limit the influence of a single station over the surrounding grid points. For

- 5 our cases, the maximum distance between every pair of closest SWS sensors is 46 km for relative humidity and 42 km for temperature; for the combined network of SWS and PWS after processing (see Sect. 4) it is 28 km for relative humidity and 21 km for temperature. These values are lower bounds of the maximum ranges in order to prevent inland grid points from having the value of the closest SWS even if they are not at the same location. For the sake of simplicity, an identical radius is chosen for temperature and relative humidity. Thus, Ffor temperature and relative humidity, SWSs distant by more than
- 10 60 km are not taken into account; this radius is set to 30 km for PWSs. The choice of 60 km instead of a 50 km radius for example is done to take into account more SWSs at each grid point (for a given inland grid point, interpolation uses 9.8 SWSs on average with a 60 km radius compared to 8.6 SWSs on average with a 50 km radius for the 26 May case). For MSLP and surface pressure, SWSs distant by more than 100 km are not taken into account; this radius is also set to 100 km for PWSs. The radius is larger for pressure because it is the minimum radius allowing to cover the entire Metropolitan France. It is due to
- 15 the small number of SWS pressure sensors and because stations with altitude higher than 750 m are discarded (see Sect. 4.4). A maximum of 10 SWSs and 30 PWSs are used at each grid point in the IDW, arbitrary limit set to diminish the program execution time.

• 3) the quality control is based on complete timeseries and it seems to be performed only in post-event phase. Can the authors consider the possibility to apply a quality flag to a certain station based on several post-event timeseries validation? this probably would pave the way also to a real time usage of these stations.

20

25

> In our current post-event quality control, we eliminate the stations flagged as bad quality stations for the complete event. We think that flagging stations is a good idea, only if the flag is updated in real-time and not only relies on several post-event time series validation. Indeed, due to constant changes in the network and in environmental conditions of PWSs, the quality of a station can be very good for a set of events but then decrease brutally without being flagged by a quality control which would not be updated in real-time, and vice versa.

See also the second major comment of referee #1: a paragraph was added in Sect. 4.4.

[revised manuscript text omitted]